# Cholinergic and noradrenergic axonal activity contains a behavioral-state signal that is coordinated across the dorsal cortex

**Lindsay Collins, John Francis, Brett Emanuel, David A McCormick\***

Institute of Neuroscience, University of Oregon, Eugene, United States

**Abstract** Fluctuations in brain and behavioral state are supported by broadly projecting neuromodulatory systems. In this study, we use mesoscale two-photon calcium imaging to examine spontaneous activity of cholinergic and noradrenergic axons in awake mice in order to determine the interaction between arousal/movement state transitions and neuromodulatory activity across the dorsal cortex at distances separated by up to 4 mm. We confirm that GCaMP6s activity within axonal projections of both basal forebrain cholinergic and locus coeruleus noradrenergic neurons track arousal, indexed as pupil diameter, and changes in behavioral engagement, as reflected by bouts of whisker movement and/or locomotion. The broad coordination in activity between even distant axonal segments indicates that both of these systems can communicate, in part, through a global signal, especially in relation to changes in behavioral state. In addition to this broadly coordinated activity, we also find evidence that a subpopulation of both cholinergic and noradrenergic axons may exhibit heterogeneity in activity that appears to be independent of our measures of behavioral state. By monitoring the activity of cholinergic interneurons in the cortex, we found that a subpopulation of these cells also exhibit state-dependent (arousal/movement) activity. These results demonstrate that cholinergic and noradrenergic systems provide a prominent and broadly synchronized signal related to behavioral state, and therefore may contribute to state-dependent cortical activity and excitability.

**\*For correspondence:**
davidmc@uoregon.edu

**Competing interest:** The authors declare that no competing interests exist.

## Editor's evaluation

This study uses behavioral monitoring and cutting-edge calcium imaging approaches to track the activity of cholinergic and noradrenergic axons in the cortex of head-fixed mice, and correlate activity with behavioral state. The authors provide compelling evidence that behaviorally related signals are broadly broadcasted to the dorsal cortex and that there is also heterogeneity across axons and areas, independent of artifacts associated with these difficult measurements.

## Introduction

Rapid, moment to moment, fluctuations in arousal state occur across species, from mice to humans, and have a significant impact on both cortical/subcortical activity and the ability to perform behavioral tasks (reviewed in *Flavell et al., 2022*; *Lee and Dan, 2012*; *McCormick et al., 2020*). In mice, rapid behavioral state fluctuations are coupled with broad changes in cortical excitation and are often accompanied by changes in arousal (pupil diameter) or movement (locomotion or orofacial movement) (*McGinley et al., 2015a*; *McGinley et al., 2015b*; *Musall et al., 2019*; *Salkoff et al., 2020*; *Stringer et al., 2019*).

The neuromodulators acetylcholine (ACh) and noradrenaline (NA) are believed to contribute to these cortex-wide fluctuations in neural dynamics by sending signals via broadly projecting axonal pathways (*Chandler and Waterhouse, 2012*; *Kim et al., 2016*; *Mechawar et al., 2000*; *Mobley and Greengard, 1985*; *Schwarz and Luo, 2015*). The release of either ACh or NA can significantly impact the activity of single cortical neurons and how these neurons interact in local and long-range networks. For example, ACh can depolarize cortical neurons by binding to both metabotropic musca-rinic and ionotropic nicotinic receptors, thereby reducing slow oscillations and facilitating higher-frequency rhythms, hallmarks of increased arousal (*Dasgupta et al., 2018*; *Hedrick and Waters, 2015*; *Metherate et al., 1992*). Similarly, NA suppresses slow oscillatory activity in the cortex via metabotropic receptors (*Sara and Bouret, 2012*). Activity patterns of cholinergic and noradrenergic neurons are highly correlated with arousal and behavioral state fluctuations. For example, cortical ACh activity increases during bouts of active whisking (*Eggermann et al., 2014*), as well as during running and licking (*Harrison et al., 2016*; *Reimer et al., 2016*), and optogenetic activation of either NA neurons in the locus coeruleus (LC) or ACh neurons in the basal forebrain (BF) promotes wakeful-ness and locomotor activity (*Carter et al., 2010*; *Irmak and de Lecea, 2014*). Moreover, both cortical ACh and NA activity in primary auditory and visual cortices track pupil dilation/constriction (*Larsen et al., 2018*; *Nelson and Mooney, 2016*; *Reimer et al., 2016*). Such studies provide clear evidence that both ACh and NA are involved in the modulation of arousal state and cortical neuronal/network dynamics.

In addition to the broad arousal-based neuromodulation described above, anatomical and func-tional evidence suggests that cholinergic and noradrenergic systems also provide more regionally specific signaling. The basal forebrain-derived cholinergic (BF-ACh) system is topographically orga-nized such that subregions within the BF preferentially project to different cortical and subcortical targets (*Gielow and Zaborszky, 2017*; *Huppé-Gourgues et al., 2018*; *Zaborszky et al., 2015*), and activation of distinct regions of the BF results in desynchronization of restricted cortical regions (*Kim et al., 2016*). Recent recordings from cholinergic neurons in different regions of the BF reveal a common signal strongly related to behavioral state, but also some degree of heterogeneity (*Robert et al., 2021*).

In addition to BF-derived cholinergic signaling, the cortex also receives input from local cortical VIP-expressing interneurons (VCINs) that release ACh, GABA, or both, depending on their specific presynaptic release machinery (*Granger et al., 2020*). While many superficial layer VIP interneurons exhibit activity related to arousal/movement (*Dipoppa et al., 2018*; *Fu et al., 2014*; *Munoz et al., 2017*; *Petersen, 2019*; *Poulet and Crochet, 2018*; *Reimer et al., 2014*; *Yu et al., 2019*), it is unclear how homogeneous or heterogeneous the activity of cholinergic VCINs is in association with arousal or behavioral (e.g. movement) state.

Within the locus coeruleus-derived noradrenergic (LC-NA) system, synchronized activity between LC neurons is often observed in relation to increases in behavioral engagement/vigilance to external events or stimuli (reviewed in *Aston-Jones and Cohen, 2005*; *Poe et al., 2020*; *Sara, 2009*), while modularity is strongly suggested by evidence of topographically distinct projection patterns (*Chandler et al., 2014*; *Kebschull et al., 2016*; *Loughlin et al., 1986*; *Loughlin et al., 1982*; *Waterhouse et al., 1983*) and paired recordings in the rat LC which can show little synchrony in spontaneous or evoked activity, at least under conditions of anesthesia (*Noei et al., 2022*; *Totah et al., 2018*). Recent inves-tigations in awake, behaving mice reveal some level of heterogeneity in neuronal activity between different LC neurons in relation to behaviorally relevant components on learned and rewarded tasks (*Breton-Provencher et al., 2022*; *Uematsu et al., 2017*).

Together, evidence suggests that a broadly synchronized, arousal and/or behavioral state-dependent component of cholinergic and noradrenergic activity may operate alongside a more local, task-specific component. Here, through simultaneous imaging of GCaMP6s expressed in multiple ACh or NA axons (and local cholinergic interneurons) across the dorsal cortex, we find that, during spontaneous behavior, cholinergic and noradrenergic signaling is dominated by a global (cortex-wide) component that is strongly predicted by arousal-related behavior.

## Results

## Relationship between neuromodulatory activity and behavioral state

To investigate the relationship between cholinergic (ACh) and noradrenergic (NA) dorsal cortical axon activity and behavioral state, we used two-photon mesoscope imaging to record GCaMP6s fluorescence in BF (basal forebrain)-ACh and LC (locus coeruleus)-NA axons in the cortex while recording facial movements, pupil diameter, and locomotion on a running wheel (*Figure 1A*) in awake, head-fixed and habituated mice. To image BF-ACh axons, injections of an axon-targeted GCaMP6s calcium sensor were made into the BF (N = 5 mice, n = 397 axon segments; see 'Materials and methods'; *Figure 1—figure supplement 1*). To image NA axons originating from the LC (*Giustino and Maren, 2018*), a cre-dependent, transgenically encoded GCaMP6s calcium reporter line was crossed with a DBH-cre line (N = 6 mice, n = 283 axon segments; see 'Materials and methods'; *Figure 1—figure supplement 1*).

To achieve a wide range of behavioral states, we habituated mice to the experimental set-up over 1–3 sessions such that the state of the animal varied from quiet resting in which movement was limited to occasional whisker twitches (this state is typically associated with smaller to medium-sized pupil diameters), to frequent, but non-continuous, movements of the whiskers/face (typically associated with moderately large pupil diameters), to prolonged (seconds) bouts of walking/running (typically associated with large pupil diameters; *McGinley et al., 2015a*; *Reimer et al., 2014*). Our habituated mice typically spent the majority of their time awake (see 'Materials and methods'), but not walking. This resting state was characterized by frequent, but brief (<0.5s) bouts of whisker 'twitches' (Animation 1), presenting as sudden movements of the whisker pad interspersed with periods without movement (*Figure 1D and F*). In addition to whisker twitching, mice also often exhibited >1 s bouts of 'whisking' characterized by large forward-backward swings of the whiskers (See Animation 1 and *Video 1*) with or without walking/running. Both whisker twitches and whisking were associated with movements of other portions of the face (e.g. snout, jaw) and increases in pupil diameter (see Figure 3). Interspersed within these periods of non-walking, the mouse occasionally would walk or run on the circular wheel for 2 s or more. These bouts of locomotion were accompanied by forward-backward movements of the whiskers, occasional eye movements (especially at the beginning of walking), and significant increases in pupil diameter (*Figure 1D and F*).

To image cholinergic or noradrenergic axon segments during variations in behavioral state, regions of interest (ROIs) were selected from across the dorsal surface of the neocortex at an imaging depth from the surface of the brain of approximately 100–200 µm (putative deep layer 1 and superficial layer 2/3). Simultaneous recordings were obtained from ROIs separated by up to approximately 3.5 mm for the BF-ACh axons and 4 mm for the LC-NA axons (*Figure 1B and C*). Selected axon segments were chosen from putative retrosplenial, somatosensory, primary and secondary motor, and visual cortices (regions defined using coarse alignment to the Allen Brain Institute Common Coordinate Framework; *Figure 1B*; *Wang et al., 2020*). Simultaneously imaged axon segments were always in the same hemisphere. Histological investigation of neurons and axons that reveal the presence of GCaMP6s and the precursors to acetylcholine (choline acetyl-transferase [ChAT]) in basal forebrain neurons or norepinephrine (tyrosine hydroxylase [TH]) in locus coeruleus cells demonstrated a strong overlap, with the vast majority of GCaMP6s (GFP+) containing neurons being ChAT+or TH+. These results confirm that our cortical axonal activity imaging is specific to these neuromodulatory systems (see *Figure 1—figure supplement 1*).

Examination of individual axon segment recordings, averages of axon segments recorded within the same ROI (50 × 50 to 400 × 400 µm), and averages of all axon segments simultaneously monitored revealed activity that closely tracked behavioral state (*Figure 1D–G*), across all dorsal cortical regions monitored (*Figure 1B*). Both cholinergic and noradrenergic axon GCaMP6s activity increased markedly in relation to whisker pad movement (twitching or whisking) and walking (*Figure 1D–G*; *Video 1*). A striking similarity was apparent between the patterns of activity in different axons, even if these axon segments were from distant recording locations. While both cholinergic and noradrenergic axons exhibited activity that was clearly related to behavioral state (as measured by whisker movements and walking), there was also an apparent difference between these two systems. GCaMP6s activity in the noradrenergic axon segments was more transient than in cholinergic axon segments during a prolonged walking bout (*Figure 1E and G*; see also *Figure 2D and E*), consistent with previous reports (see also below; *Reimer et al., 2016*).

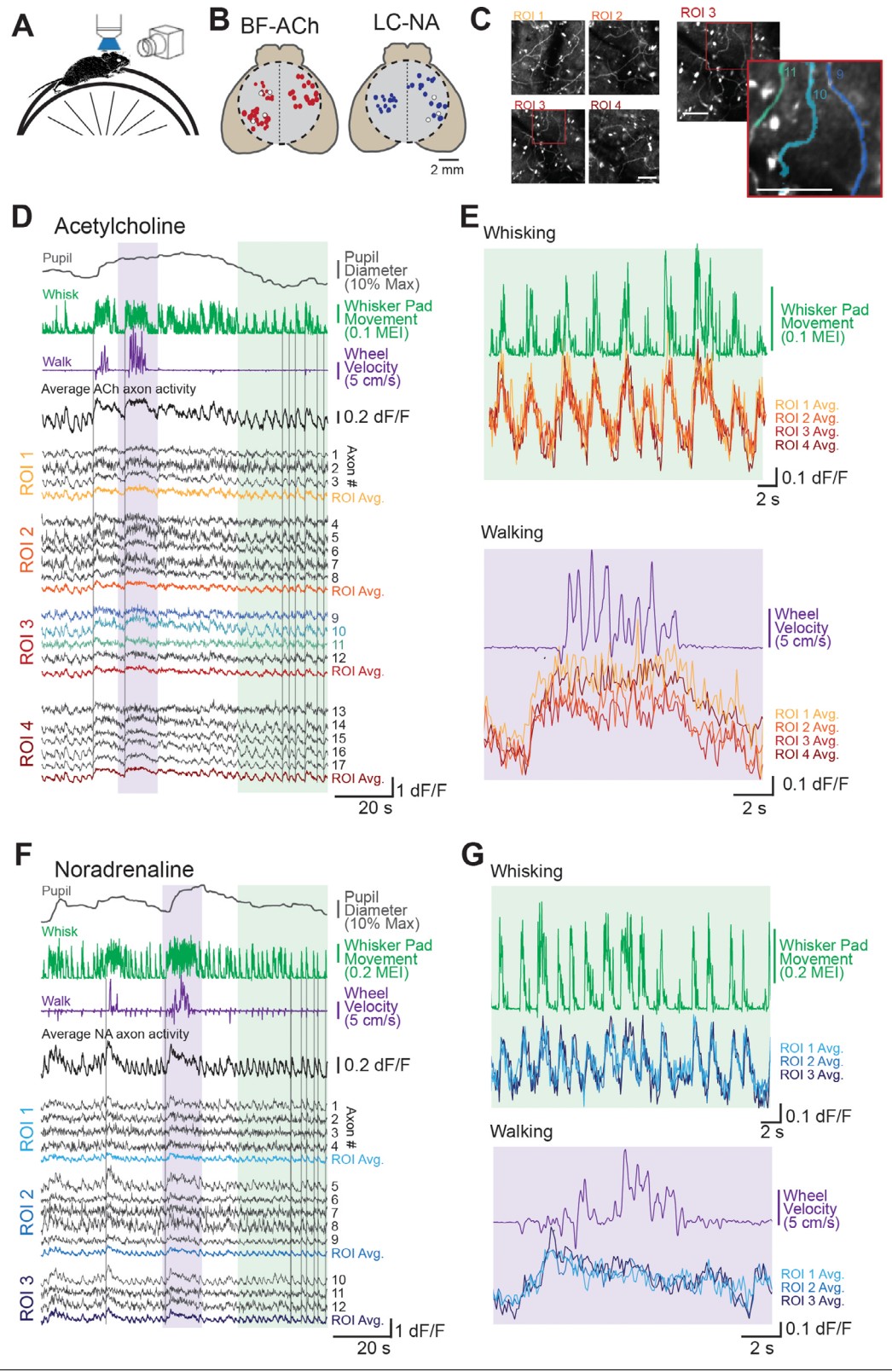

**Figure 1.** Two-photon imaging of dorsal cortical cholinergic and noradrenergic activity reveals strong state dependence.

(**A**) Schematic of experimental setup. Awake mice were placed atop a running wheel underneath a two-photon mesoscope objective for imaging of axonal activity through GCaMP6s. A second side camera was used to

*Figure 1 continued on next page*

*Figure 1 continued*

capture facial movements. (**B**) To allow for imaging across a large region of cortex, mice were implanted with an 8 mm diameter glass cranial window (dotted circle) on the dorsal brain surface. Recordings could be collected from regions of interest (ROIs) within a 5 × 5 mm field of view. Estimated locations of all ROIs are shown for all BF-ACh (left) and LC-NA (right) axon segment recordings. Size of each circle corresponds to the size of the corresponding square ROI, ranging from 50 × 50 to 400 × 400 μm. Typically three or four ROIs, always within the same hemisphere, were used for imaging. White circles represent ROIs shown in panels (**D–G**). (**C**) Left: mean projection image of four simultaneously recorded ROIs (corresponding data shown in panel **D**). Right: example of Suite2P-defined axons. Three axons are shown in the inset image, which correspond to the three blue dF/F traces in panel (**D**). Scale bars represent 50 μm. (**D–G**) Example recordings of GCaMP6s activity in 17 BF-ACh simultaneously imaged axon segments in four ROIs (**D, E**) and 13 LC-NA axon segments in three ROIs LC-NA (**F, G**). Pupil diameter, whisker pad movement (represented as motion energy index [MEI], as described in 'Materials and methods'), and wheel movement (both rotational and vertical) are shown alongside dF/F of individual axons recorded simultaneously across ROIs shown in panel (**B**). Vertical gray lines in (**D**) and (**F**) are for reference and are aligned to the onset of examples of walking or whisker movements (see Animation 1 and *Video 1*). Average activity across all axons simultaneously monitored as well as within a ROI are shown (**D, F**). Colored regions in (**D**) and (**F**) are expanded for illustration in (**E**) and (**G**). Axon GCaMP6s data in (**E**) and (**G**) are represented as ROI averages. Histological investigation revealed that GCaMP6s expression was specific for cholinergic and noradrenergic neurons and axons (see *Figure 1—figure supplement 1*).

The online version of this article includes the following figure supplement(s) for figure 1:

**Figure supplement 1.** Histological verification of GCaMP6s labeling specificity for cholinergic and noradrenergic neurons and axons.

To examine the relationship between cholinergic and noradrenergic axon activity and behavioral state transition, we aligned axonal recordings to the onsets and offsets of facial whisker pad twitches, whisking bouts, and walking bouts (*Figure 2A–C*, respectively; *Figure 2—figure supplement 1A–F*). We confirmed that any movement-related axonal responses were not affected by motion artifacts in two ways. First, we compared GCaMP6s fluorescence to non-activity-dependent autofluorescence in small blebs within the imaging field (*Figure 2A–C*, blebs; see 'Materials and methods'). Second, we compared GCaMP6s fluorescence activity to non-activity-dependent mCherry fluorescence, introduced into axons via viral AAV injection (*Figure 2F*, *Video 2*; see 'Materials and methods'). Whisker twitches were defined as whisker pad movements that lasted less than 0.5 s, while a whisking bout was defined as whisker pad movement that lasted longer than 1 s. Both were required to be preceded by a period of 1 s of non-whisker movement to be included in our analysis.

At the onset of both facial movements/twitches, as well as whisking bouts (which involve both larger and more prolonged movements of the whiskers than whisker twitches), both BF-ACh and LC-NA axonal activity sharply increased (*Figure 2A and B*; *Figure 2—figure supplement 1A–F*). Importantly, no change in fluorescence was observed at the onset of whisking in either autofluorescent blebs (*Figure 2A and B*) or non-activity-dependent mCherry axon fluorescence (*Figure 2F*, *Video 2*). Expanding the time base revealed that the onset of the increases in average BF-ACh and LC-NA axonal activity occurred, on average, nearly simultaneously with the onset of whisking (*Figure 2B*, insets). Whisker movements reached their peak prior to GCaMP6s fluorescence (*Figure 2B*, insets), presumably owing, at least in part, to the slower kinetics of intracellular calcium dynamics and the GCaMP6s fluorescence reporter mechanism (*Chen et al., 2013*) in comparison to whisker movements (see *Figure 2—figure supplement 2*). Both cholinergic and noradrenergic axon activity decreased along with whisker movement just prior to onset of whisker twitches or a bout of whisking (*Figure 2A and B*). This decrease in whisker movement and ACh/NA activity is the result of our requirement of no whisker movements for at least 1 s prior to the bout, for the bout of whisker movement to be included in our analysis (see 'Materials and methods').

At the cessation of facial movement, BF-ACh and LC-NA axon activity decreased slowly (over 1–2 s; *Figure 2B*, offset), in comparison to the rapid rise at movement onset (*Figure 2B*, onset). Previous studies of cholinergic and noradrenergic axonal GCaMP activity revealed that cholinergic fibers exhibit more sustained responses than noradrenergic axons during a bout of walking (*Reimer et al., 2016*). By examining whisking bouts of different durations, from 1 s to >5 s, we also observed that cholinergic axonal GCaMP6s responses are significantly more sustained than the noradrenergic axonal responses (*Figure 2D and E*). For example, during prolonged (>5 s) bouts of whisking, the

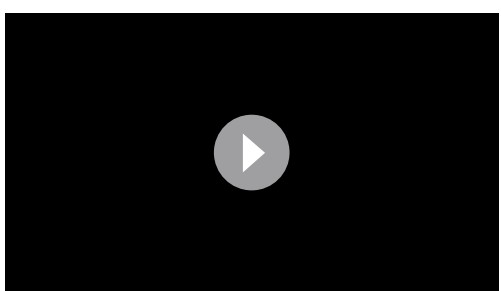

**Video 1.** Axon imaging and behavioral acquisition methodology. Related to *Figure 1*. A headpost was implanted on the skull overlying the dorsal cortex (top of video = anterior). Cortical regions coarsely aligned to the Allen Brain Institute Common Coordinate Framework are shown for reference. An 8 mm circular cranial window was placed over multiple cortical regions. In each imaging session, axons or cells within up to a 5 × 5 mm region could be imaged simultaneously. An epifluorescence image of the visible brain areas for this example session is shown along with four red squares, which indicate the position of each of four regions of interest (ROIs) recorded simultaneously. Next, all collected data are shown for this example session. Top left: position of regions of interest. Top middle: video recording of the mouse's face (used for pupil diameter and whisker pad motion extraction). Bottom left: traces of dF/F for each isolated axon shown on the right along with whisker pad motion energy, walking speed, and pupil diameter. Right: four videos of ACh axon imaging aligned in time to behavioral variables shown on the left. Videos are sped up by 10× but are otherwise unprocessed (i.e. no motion correction or temporal resampling). The pupil is large in this video owing to low UV/white light levels to enhance axon imaging SNR (see 'Materials and methods').

https://elifesciences.org/articles/81826/figures#video1

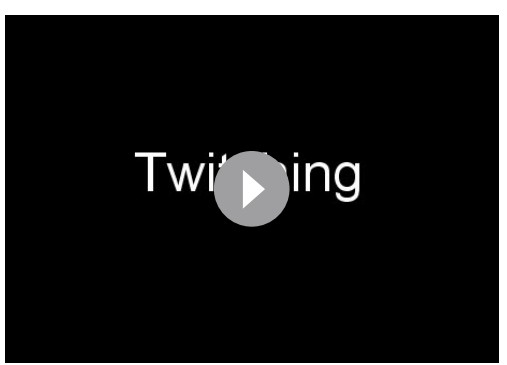

**Animation 1.** Example of mouse face during twitching, whisking, and walking. Related to *Figures 1 and 2*. Video recording of a mouse's face during periods of twitching, whisking, and walking. Note the difference between a short (< 0.5 s) twitch and a sustained (> 1 s) whisking bout. Also note the prominent whisker pad motion during bouts of walking.

NA axonal response typically exhibited a strong increase in GCaMP6s fluorescence at the beginning of the whisking bout, which decreased over the next 0.5–1.5 s to a steady value that was on average 11% of the peak value (activity at 2–3 s compared to peak, t = −15.14, p<0.0001; *Figure 2E*). In contrast, cholinergic fibers exhibited a more sustained response, such that the activity 2–3 s into the prolonged whisking bout was 93% of the peak (activity at 2–3 s compared to peak, t = 0.91, p=0.36; *Figure 2D*).

The GCaMP6s activity of both BF-ACh and LC-NA axons increased prior to the onset and slowly decreased at the offset, of a walking bout (*Figure 2C*). Again, no movement-related change in fluorescence was observed for either the autofluorescent blebs (*Figure 2C*) or non-activity-dependent mCherry in axons (*Figure 2F*, *Video 2*). To determine the uniformity of the GCaMP6s response across axons and behavioral events (e.g. *Figure 2—figure supplement 1*), we calculated the probability that a given axon would respond at the onset of a behavioral event (twitch, whisk, or walk). We defined 'responsive' as a significant increase in dF/F between 1 s following event onset and 1 s prior to event onset (*Figure 2—figure supplement 1G and H*; Student's *t*-test, alpha = 0.01), in comparison to a distribution obtained from shuffled data (*Figure 2—figure supplement 1I and J*). We found that a large number of BF-ACh and LC-NA axons increased activity at the onset of twitching (BF-ACh: 44.2 ± 30.9%; LC-NA: 47.3 ± 28.4%) and whisking (BF-ACh: 78.8 ± 23.5%; LC-NA: 67.3 ± 30.9%) bouts. On average, each BF-ACh axon significantly increased activity on 43.5 ± 15.7% of twitching and 75.8 ± 25.0% of whisking bouts. Similarly, on average, each LC-NA axon segment significantly increased activity on 46.7 ± 13.7% of twitching and 74.4 ± 24.4% of whisking bouts. These results indicate that, on average, with each whisker twitch, approximately half of NA and ACh axons will show increased activity responses, while with the transition to full whisking ~75% of NA and ACh axons are activated. At the onset of whisking bouts, simultaneously recorded axons were varied in their response even for whisking bouts that exhibited very similar changes in average axonal dF/F (*Figure 2—figure supplement 1K–O*).

Walking bouts, which are associated with strong increases in cholinergic and noradrenergic activity (e.g. *Figures 1 and 2C*, *Figure 2—figure supplement 1C and F*) were not associated with a significant difference 1 s after walk onset in

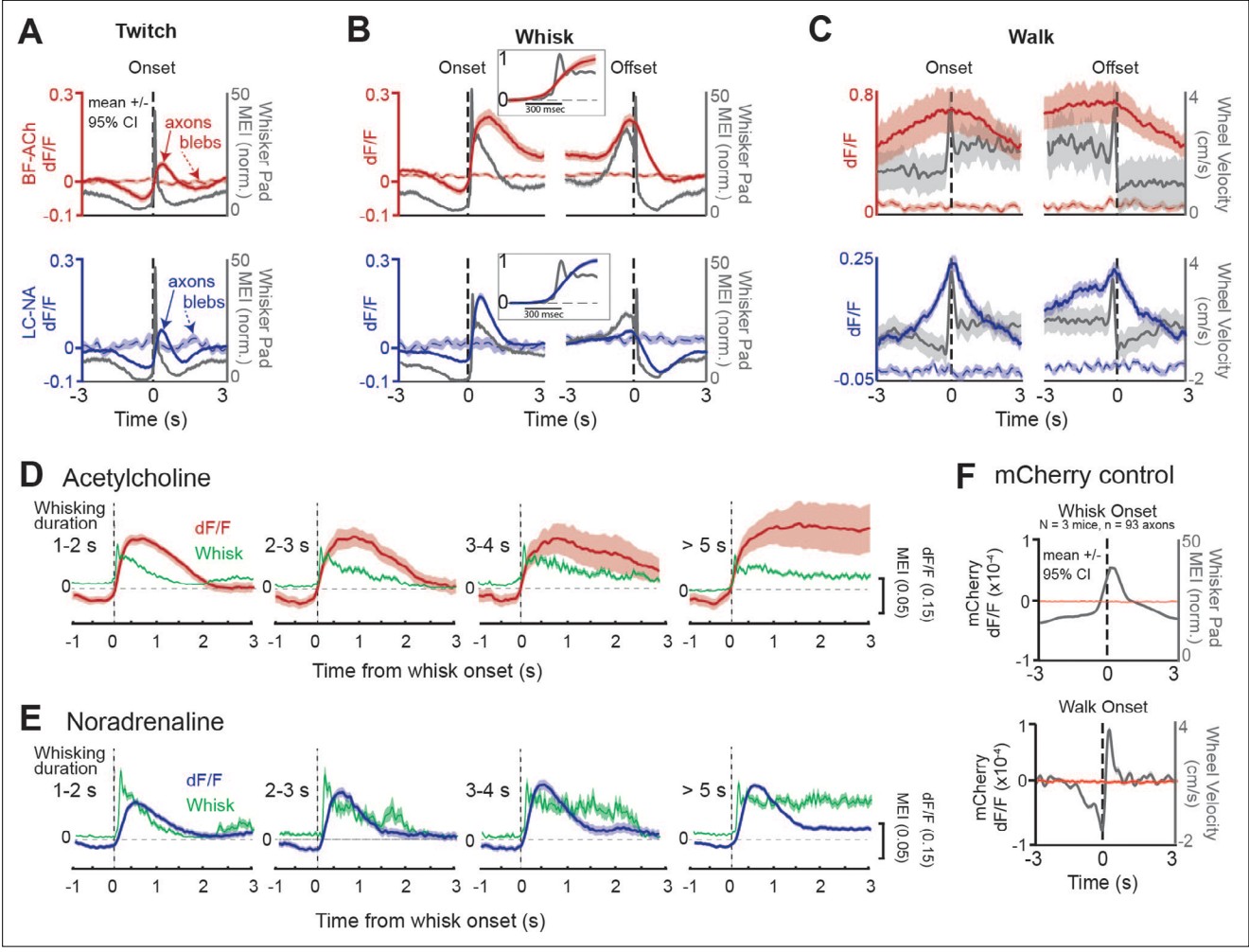

**Figure 2.** Basal forebrain-derived cholinergic (BF-ACh) and locus coeruleus-derived noradrenergic (LC-NA) activity track behavioral state transitions. (**A–C**) Mean GCaMP6s fluorescence in BF-ACh (top) and LC-NA (bottom) axons aligned to the onset and offset of whisker twitching (**A**), a bout of full whisking (**B**), and walking (**C**). Gray traces represent averaged whisker pad motion energy (**A, B**) or walking speed (**C**). Shaded region represents a 95% confidence interval. Fluorescence of nearby blebs (dotted lines) was not affected by whisking or walking. (**B**), insets. Expanded time base, with data normalized from 0 to 1, for comparison of onset kinetics of average whisker movements and cholinergic/noradrenergic axon activity. (**D, E**) Activation of cholinergic (**D**) and noradrenergic (**E**) axons during whisking bouts of varying durations, from 1 to 2, 2–3, 3–4, and >5 s. Note that the cholinergic (red traces) axonal responses maintain activity during prolonged whisking (green traces), while the noradrenergic axonal responses (blue traces) are more transient. Shaded region represents a 95% confidence interval. For all panels, ACh data obtained from N = 5 mice, n = 397 axon segments over 1108 twitches, 330 whisking onsets, 201 whisking offsets, and 101 walking bouts. NA data obtained from N = 6 mice, n = 283 axon segments over 644 twitches, 400 whisking onsets, 353 whisking offsets, and 34 walking bouts. (**F**) Mean non-activity-dependent mCherry fluorescence (red traces) in axons aligned to the onset of whisking (top) and walking (bottom). Gray traces represent whisker pad motion energy (top) and walking speed (bottom). Note that the dF/F of mCherry axons is three orders of magnitude smaller than that of GcaMP6s axons and does not fluctuate at the onset of walking or whisking. For mCherry control experiments, 93 axon segments were imaged in three mice during 147 whisker twitches, 564 whisking onsets, 529 whisking offsets, and 105 walking bouts.

The online version of this article includes the following figure supplement(s) for figure 2:

**Figure supplement 1.** Individual axonal responses to behavioral state changes.

**Figure supplement 2.** Distribution of power and coherence for pupil, cholinergic, noradrenergic, whisking, and VIP-expressing interneuron (VCIN) activity.

**Figure supplement 3.** Variable timing of onset/offset of whisking and walking and fluorescence of cholinergic and noradrenergic axons and VIP-expressing interneurons (VCINs) during different states.

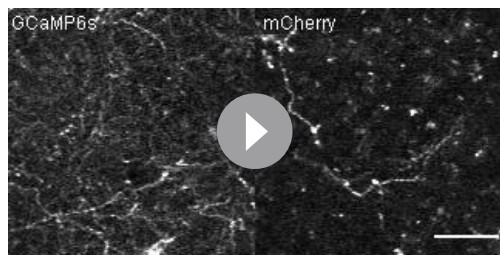

**Video 2.** Comparison of changes in fluorescence with whisker movements in GCaMP6s axons versus no discernible changes in fluorescence in non-activity-dependent mCherry axons. Scale bar = 50 μm. Related to *Figures 1 and 2*. These two examples are representative recordings obtained from two different animals, while the animals were performing un-instructed whisker twitches and whisking (not shown). https://elifesciences.org/articles/81826/figures#video2

comparison to 1 s prior to walk onset, owing to an increase in average cholinergic and noradren-ergic axon activity in the 1 s prior to walk onset (see *Figure 2C*, *Figure 2—figure supplement 1C and F*).

We suspected that the strong activation of ACh and NA axons prior to the onset of walking was the result of increases in movement (e.g. whisking) and arousal (e.g. pupil diameter) during this time period. To test this hypothesis, we compared the onset/offset of whisker pad move-ment and walking. We observed that whisker pad movement is consistently initiated, with a vari-able onset, 0.5–5 s prior to walking initiation and typically persists for approximately 1–3 s, with a variable offset, after the cessation of walking (*Figure 2—figure supplement 3A and B*). Thus, the lack of tight temporal correlation between onset/offset of walking with cholinergic and noradrenergic activity is likely the result of a vari-able timing of onset/offset of whisking/arousal in relationship to walking bouts, and the strong asso-ciation between ACh and NA axonal activity and behavioral state (which is tightly linked to whisker movements; *Figure 1E and G*).

Since we observed a strong relationship between whisker pad motion energy and modulatory axon activity, we wondered whether axon activity was linked to whisker movements per se or to facial movement more generally. To answer this question, we compared the relationship between axon activity and motion of the whisker pad to the relationship between axon activity and motion of the mouth, snout, and full face in a subset of recordings comprising 65 BF-ACh axons and 50 LC-NA axons (*Figure 3A*). Overall, these variables were highly correlated as motion in one part of the face often accompanies motion in other regions of the face. However, we were able to identify occasional instances where facial motion was predominantly localized to the whisker pad, jaw, or snout regions (green, red, and yellow bars, respectively, in *Figure 3B*) in all analyzed recording sessions.

First, we investigated the relationship between axonal activity and facial movements by calculating the coherence between GCaMP6s dF/F and movement within distinct regions of the face (jaw, snout, whisker pad) as well as movement across the full face. Previous studies have demonstrated a strong coherence at low frequencies (<3 Hz) between cholinergic or noradrenergic cortical axonal activity and behavioral state, as measured by changes in pupil diameter (*Harrison et al., 2016*; *Larsen et al., 2018*; *Nelson and Mooney, 2016*; *Reimer et al., 2016*; *Robert et al., 2021*; *Sturgill et al., 2020*). In similarity to these studies, we found that coherence between both BF-ACh and LC-NA axon activity and facial movement and pupil diameter was highest at low frequencies (<1 Hz; *Figure 3C*). However, in comparison to walking speed and pupil dilation, we observed a stronger coherence between both BF-ACh and LC-NA axon activity and facial movements (including snout, jaw, whisker pad, and full face motion energy; *Figure 3C and D*), suggesting that facial movement is tightly coupled to changes in cholinergic and noradrenergic signaling in the cortex.

The large drop in coherence between axonal activity and measures of behavioral state at frequen-cies above approximately 1 Hz likely represents more of a limitation of our imaging method to detect axonal activity at frequencies above 1–2 Hz, than a true lack of coherence (see *Figure 4A–E* and 'Materials and methods'; *Figure 2—figure supplement 2*). Indeed, examining the distribution of power in our GCaMPs signal from cholinergic and noradrenergic axons reveals a strong drop off above approximately 1 Hz, such that only low power is observed at frequencies above approximately 2–4 Hz (*Figure 2—figure supplement 2*). Thus, our study of comparing activity in cholinergic and noradrenergic axons across the cortex with behavioral state is most relevant to low frequency (≤1 Hz) events, similar to previous investigations (*Reimer et al., 2014*).

The temporal relationship between BF-ACh activity and facial movement, as measured by cross-correlation, was closely time-locked such that axonal activity was observed very shortly (within 100–200

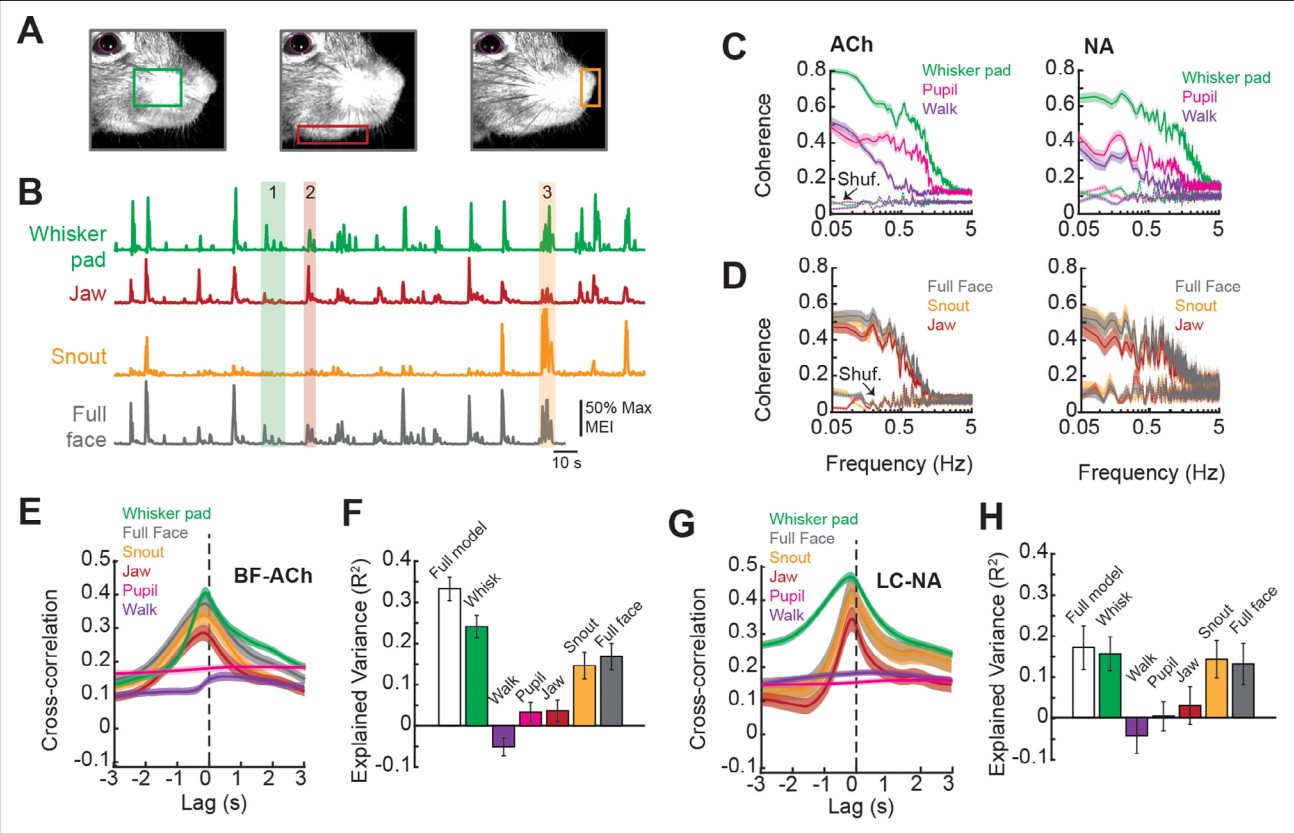

**Figure 3.** Facial movements are highly predictive of basal forebrain-derived cholinergic (BF-ACh) and locus coeruleus-derived noradrenergic (LC-NA) axon activity.

(A) Restricted regions within videos of the face were analyzed to capture movement of the (left to right) whisker pad, jaw, and snout. (B) Example traces of the motion energy index (MEI) recorded from the whisker pad, jaw, snout, and across the entire frame (full face). Instances where motion energy of the face was predominated by whisker pad (1), jaw (2), or snout (3) movement are noted by green, red, and yellow bars, respectively. (C) Coherence between average BF-ACh or LC-NA axon activity and movement of the whisker pad, walking, and pupil diameter. All simultaneously imaged axon segments within a session were averaged prior to calculation of coherence. (D) Coherence between average BF-ACh or LC-NA axon activity and motion energy in the full face, snout, and jaw. It should be noted that examining coherence between adjacent segments of the same axon reveal our imaging method to be unable to accurately represent variations in activity above approximately 1 Hz, indicating a frequency limitation in the present coherence measures (see *Figure 4* and *Figure 2—figure supplement 2*). (E) Cross-correlation between arousal measures and BF-ACh axon activity. (F) Amount of variance in BF-ACh axon activity that is explained by pupil, whisker movements, walking, snout movements, jaw movements, and movement energy in the full face view. (G, H) Same as (E, F) for LC-NA axons. Error represents a 95% confidence interval for panels (C–H). Results presented in (E–H) were obtained after low-pass filtering signals at 1 Hz. For all panels, ACh data taken from N = 3 mice, n = 65 axon segments and NA data taken from N = 3 mice, n = 50 axon segments.

The online version of this article includes the following figure supplement(s) for figure 3:

**Figure supplement 1.** Quantification of linear regression predictive error.

ms) following the onset of facial movements (*Figure 3E*). This delay is approximately the same as the rise-time required for GCaMP6s to bind $Ca^{2+}$ and fluoresce in response to the initiation of action potentials (100–150 ms; *Chen et al., 2013*), indicating that the behavioral state-dependent onset of facial movements and increased activity in the BF-ACh system occurred within a relatively short time period, beyond our ability to resolve (see also *Figure 2B* insets).

We determined the predictive power of each of the six behavioral arousal measures (pupil diameter, walking velocity, and motion energy of the whisker pad, snout, jaw, and full face) on axon activity using ridge regressions to avoid overestimation resulting from multicollinearity of predictor variables since arousal state variables are not independent. The full model accounted for 33.3 ± 2.9% of variance in BF-ACh axonal activity (*Figure 3F*). Movement of the whisker pad area explained more of the variance in BF-ACh axonal activity than any other variable (whisker pad movement explained variance = 24.1 ± 2.8%; F = 89.2, p<0.0001; one-way ANOVA). Walking speed explained less variance

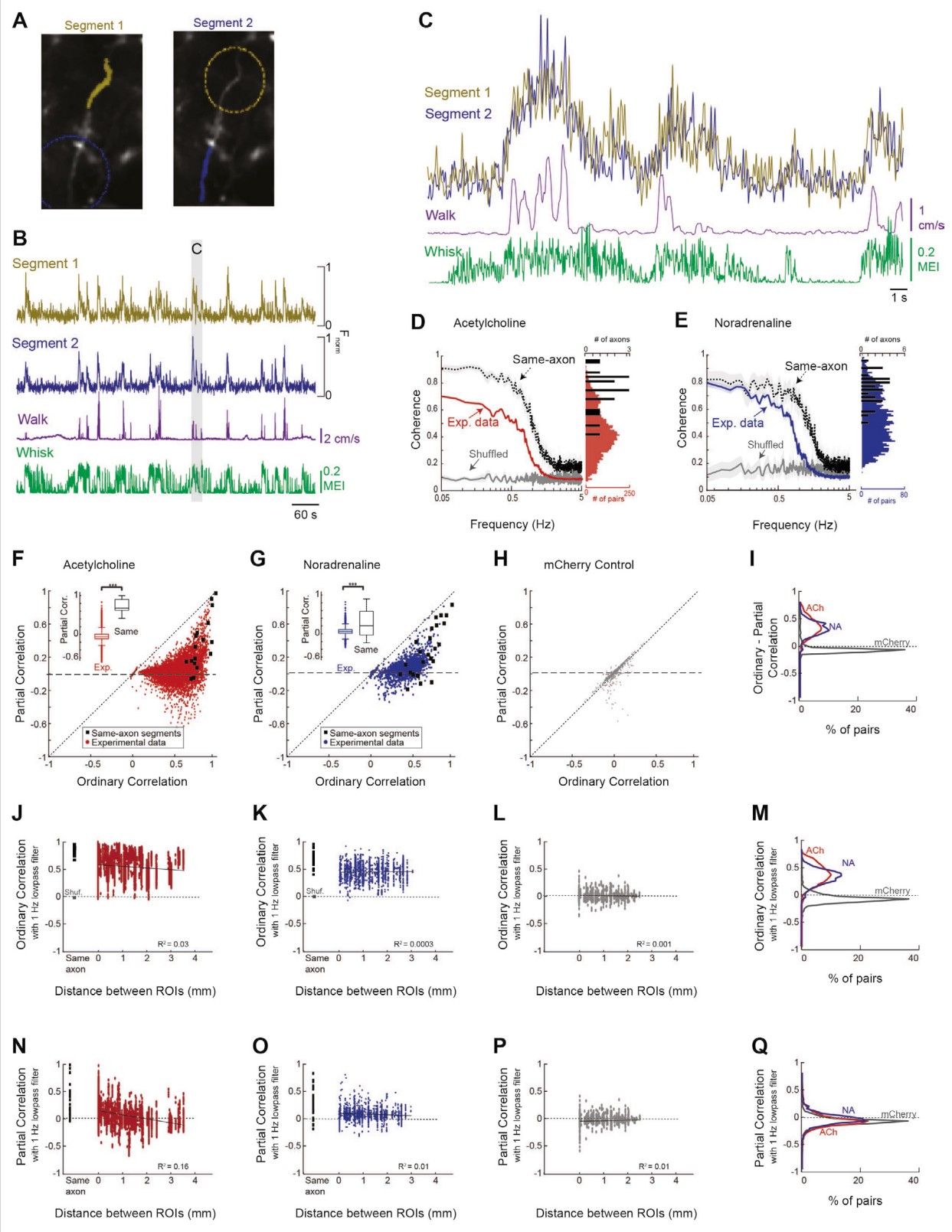

**Figure 4.** Cholinergic and noradrenergic axon segments across the dorsal cortex exhibit a common signal related to behavioral state. (**A**) To examine the relationship between monitored axon segments, we first measured the relationship between GCaMP6s Ca$^{2+}$ fluorescence signals measured in adjacent segments of the same axon (e.g. yellow labeled segment 1 vs. blue labeled segment 2; dotted circles indicate adjacent segments). Theoretically, all, or nearly all, action potentials will invade adjacent segments of the axon, providing a measure of the limitations of the frequency of

*Figure 4 continued on next page*

*Figure 4 continued*

activity that we can monitor with our methodology. (**B**) Plot of the GCaMP6s activity observed in the two adjacent segments of the same axon in relation to walking and whisker motion energy. (**C**) Expanded example of the measured activity in the two connected axon segments for detail. Note that the signal between adjacent segments of the same axon is coherent at low frequencies, but not at higher frequencies. (**D**) Coherence between segments of the same axon segments (black trace; N = 5 mice, n = 18 same-axon segment pairs) exhibit a high coherence at frequencies below approximately 1 Hz and a strong decrease at frequencies between 1 and 3 Hz, indicating that our monitoring methodology is limited to frequencies of approximately <1–3 Hz. Comparison of coherence between all axon segments simultaneously monitored (N = 5 mice, n = 7272 pairs of axons) reveal a strong average coherence at frequencies less than 1 Hz, but this average coherence is lower than that for axon segments of the same axon. Plotting the distribution of average coherence at frequencies between 0.05 and 0.5 Hz reveals a broader distribution for heterogeneous axon segments in comparison to segments of the same axon. (**E**) Comparison of coherence for adjacent segments of the same axon and all axon segments for noradrenergic axons (N = 4 mice, n = 30 same-axon segment pairs, n = 1699 different axon pairs). Coherence is plotted as average +/- 95% confidence intervals. Plotting the distribution of low-frequency coherence reveals a broader distribution for heterogeneous axon segments in comparison to segments of the same axon. (**F, G**) Plot of partial correlation versus ordinary correlation for cholinergic (**F**) and noradrenergic (**G**) axons reveals a strong contribution of a common signal to the correlations between most axon segments. The common signal was calculated as the average signal between all axon segments simultaneously imaged, excluding the two axon segments for which the partial correlation was calculated. Black dots represent segments of the same axon, which should exhibit a correlated signal that is independent of the common, group-average signal. (**H**) Plot of partial correlation versus ordinary correlation for non-activity-dependent mCherry labeled axons. (**I**) Distribution of difference between ordinary and partial correlation for acetylcholine (ACh), noradrenaline (NA), and mCherry axon segments. (**J, K**) The correlation in GCaMP6s signal between cholinergic (**F**) and noradrenergic (**G**) axon segments (signals low-pass filtered at 1 Hz) versus distance between the regions of interest containing the axon segments. 0 mm indicates axon segments within the same region of interest. Black squares represent correlation between adjacent segments of the same axon. Red or blue dots illustrate the correlations between activity in all axon segments, excluding those that were clearly part of the same axon. Note that for both ACh and NA axons, there is generally a high correlation if the axon segments are adjacent and from the same axon (black squares). For all axon segments, the correlations are much more varied, ranging from around 0 to greater than 0.8. These correlations change only slightly with distance, even up to 4 mm, indicating that there is broad synchrony of cholinergic and noradrenergic axonal activity across the cortex. (**L**) Distribution of correlations with distance between non-activity-dependent mCherry axonal fluorescence. (**M**) Distribution of ordinary correlation between segments of ACh, NA, and mCherry axons. (**N, O**) Dependence of partial (residual) correlations between ACh (**N**) and NA (**O**) axon segments with distance. (**P**) Partial correlations plotted against distance between segments for mCherry-labeled axon segments. (**Q**) Plot of distribution of partial correlations between ACh, NA, and mCherry axon segments.

The online version of this article includes the following figure supplement(s) for figure 4:

**Figure supplement 1.** Features of axonal imaging data relevant to interpreting imaging results.

in axon activity than any other behavioral measure ($R^2$ = –0.05 ± 0.02). Movements of the snout were also informative, as were motion energy in the full face video (*Figure 3F*). It should be noted that behavioral measures such a pupil diameter or walking speed, which exhibit a significant coherence with BF-ACh and NA-LC activity at low frequencies (1 Hz and 0.1 Hz, respectively), may not explain a high percentage of variance of the total axonal activity, presumably owing to low coherence at higher frequencies (see *Figure 3C and D*). Whisking can explain significantly more axonal fluorescence variance (versus walking or pupil diameter) since whisker movements are more highly coherent, and across a broader range of frequencies, with axonal activity than our other measures (*Figure 3C*).

Similar to BF-ACh, activity of LC-NA axons lagged facial movements by 100–200 ms and facial (whisker, snout, full face) movements explained more variance in the noradrenergic axonal signal than walking speed or pupil diameter (*Figure 3G and H*; *F* = 17.16, p<0.0001). Although we used ridge regressions to avoid overestimation, there still is the possibility that the regression overestimated the relationship between axon activity and behavior. We used the 'session permutation' method (*Harris, 2020*) to address this concern. Briefly, this method determines the likelihood of a correlation arising from truly non-correlated data by comparing the prediction error obtained from regressions of matching (simultaneously recorded) sessions, to the distribution of prediction error for regressions of all permutations of unmatched sessions. For both BF-ACh and LC-NA data sets, prediction error for matched whisking and walking data sets was low in comparison to the distribution of unmatched sessions; therefore, we conclude that the relationships determined by linear regressions are valid (*Figure 3—figure supplement 1*).

## Evidence for global vs. heterogeneous activity within ACh and NA axon segments across the dorsal cortex

To examine the possibility of both common and unique signals within cholinergic and noradrenergic axon segments across the cortex, we sought to measure the coherence and correlation in GCaMP6s

signal between axon segments of the same modulatory system, and the ability of a common signal to explain this activity, as a function of distance and behavioral state (*Figure 4*). To establish an interpretable baseline for these measures, we compared the results obtained in all axon segments with those obtained by comparing the activity in a random sample of sub-segments of the same axons (n = 18 ACh axons; n=30 NA axons; *Figure 4*). This control allowed us to determine to what extent cholinergic and noradrenergic axons act homogeneously (as if they are segments of the same axon).

## Cholinergic and noradrenergic activity contains a global signal related to behavioral state

To determine the extent to which cholinergic and noradrenergic axonal activity was similar across the cortex, we first calculated the correlation between individual axons recorded in the same imaging session across multiple ROIs. Overall, we found a high average correlation between GCaMP6s activity in both cholinergic and noradrenergic axon segments (ACh: 0.54 ± 0.19; NA: 0.44 ± 0.15). However, we also observed a broad distribution of correlations that slightly decreased (for ACh axon segments), with distance, suggesting that heterogeneous signals may also be present (described below). Comparing the average coherence between all cholinergic/noradrenergic axon segments revealed a high coherence at frequencies less than approximately 1 Hz, and this group average coherence decreased markedly at frequencies above approximately 1 Hz (*Figure 4D and E*, Exp. data). This high level of average coherence between simultaneously imaged cholinergic/noradrenergic axon segments at frequencies <1 Hz suggests the presence of a strong common signal that is broadly expressed across these segments (*Figure 4D and E*). The drop off in coherence at frequencies >1 Hz is most likely a consequence of the frequency limitations of our data collection technique (see below and *Figure 2—figure supplement 2*).

The presence of strong coherence and correlation between ACh or NA axon activity and behavioral state (e.g. whisker movements) suggests that coherence/correlations between axon segments may arise simply because they are both coherent/correlated to a common signal, such as movements/arousal. To examine the possibility of such a signal driving the correlation between axonal segments, we calculated a 'common signal' as the average signal for all simultaneously imaged axon segments, minus the two axon segments under consideration (i.e. being correlated). This average or 'common signal' within cholinergic or noradrenergic axons is strongly coherent with movements of the whisker pad, especially at lower frequencies (*Figure 3C and D*). Next, we calculated the partial correlation between each pair of simultaneously imaged axon segments, given the average of all other simultaneously imaged axon segments (i.e. the common signal, see 'Materials and methods').

Plotting the residual partial correlation versus the ordinary (full) correlation revealed a large decrease in correlation for the vast majority of axon segments when this common signal is taken into account, such that the partial correlation centered around approximately 0 (ACh: mean = 0.03 ± 0.14 [SD]; NA: 0.07 ± 0.11; *Figure 4F and G*). This result indicates that much of the ordinary correlation between axon segments, particularly at values below approximately 0.5–0.6, may have resulted from the correlation of the two axon segments to a common signal related to behavioral state.

At correlation values above 0.5–0.6, there are pairs of axon segments whose partial correlation rises toward 1 as the ordinary correlation also rises (*Figure 4F and G*). A similar phenomenon occurs in correlating axon segments from the same axon, indicating that this may result from imaging branches of the same axon (however, see below – distant cell bodies of cortical cholinergic interneurons can also exhibit this property). Examining the fluorescence of non-activity-dependent mCherry-labeled axons did not reveal a significant decrease with partial (vs. ordinary) correlation, indicating that the effect observed in cholinergic and noradrenergic axons with this comparison is not due to movement-artifacts (*Figure 4H*).

What is the source of the common signal between axon segments? One possibility is that variations in behavioral state correlated with animal movement may represent a significant portion of the common signal. To test this hypothesis, we examined the degree to which restricting our analysis to periods in which behavioral state was less variant reduced the ability of the effects of the common signal on inter-axonal correlations. Specifically, we compared the difference between partial and ordinary correlation values for (1) full data set; (2) periods that exclude walking, but may include whisking; and (3) periods that exclude both walking or whisking. We found that removing periods of both walking and whisking significantly (p<0.01) lessened the difference between the partial and ordinary

correlations, while removing periods of walking did not have a significant effect (*Figure 4—figure supplement 1J*). This result confirms that a significant portion of the common signal in ACh and NA axon activity results from behavioral state variations associated with movements.

## Characteristics of axonal segment activity that is not part of the common signal

Axons can signal heterogeneous activity either through their own trial-to-trial variations in activity or through their variations in correlation with other modulatory axons. To test these two hypotheses, we examined the correlation/coherence between activity in heterogeneous axon segments after accounting for the common signal (and in comparison to same axon activity and non-activity-dependent mCherry axons), as well as the trial to trial variation in activity in the simultaneously imaged population of axon segments.

While the average coherence between axon segments was high, there was a broad distribution of coherence between simultaneously monitored axon segments, as revealed by plotting the distribution of average coherence at frequencies between 0.05 and 0.5 Hz for all simultaneously recorded axon segments. The distribution of coherence values was substantially broader than that observed in pairs of segments from the same axon (*Figure 4D and E*). The broad range of relationships between activity in simultaneously recorded axon segments is also illustrated by examining the correlation of fluorescence between these segments (*Figure 4F and G*). Comparison of axonal fluorescence correlations for both cholinergic (*Figure 4F*) and noradrenergic (*Figure 4G*) axons revealed a broad distribution, ranging from –0.03 to 0.99 for ACh and –0.08 to 0.91 for NA, with an average correlation of 0.54 ± 0.19 for ACh and 0.44 ± 0.15 for NA. The average correlations between ACh axon segments decreased weakly with increased distance between segments (*Figure 4J*), with the correlation between inter-axonal segment activity correlation and distance between segments being –0.12 (p<0.001). In contrast, there was no significant relationship between distance and correlation between NA axon segments ($r = 0.02$; p=0.46; *Figure 4K*). As with coherence, the distribution of correlations between both cholinergic and noradrenergic axons was substantially broader than that observed in pairs of segments from the same axon (*Figure 4F, G, J and K*).

We addressed whether or not the broad distribution of correlations between heterogeneous axons was the result of variations in their correlation with the common signal by examining the distribution of partial (residual) correlations between these axon segments, once the common signal was taken into account (*Figure 4N–Q*). Interestingly, the distributions of residual correlations between heterogeneous cholinergic, noradrenergic, or non-activity-dependent mCherry axons were nearly identical (*Figure 4Q*). We found no statistical differences between these distributions (p>0.001; *F* = 76.98; one-way ANOVA). However, comparing the distribution of partial (residual) correlations between cholinergic axons revealed a significant distance dependence ($r = -0.4$; p<0.01), such that axon segments within the same ROI exhibited a higher correlation than axon segments that were more distant (*Figure 4N*). This result may be explained by the axon segments within the same ROI being segments of the same axon. Indeed, examining the distribution of partial correlations in axon segments from the same axon revealed a similar positive-biased distribution (*Figure 4N*). In noradrenergic axon segments, the partial (residual) correlation decreased slightly with distance, dropping from 0.09 ± 0.12 within the same ROI, to 0.06 ± 0.10 between all other axon segment pairs (*Figure 4O*). The overall correlation between distance and residual partial correlation for NA axon segments was very low, but statistically significant ($r = -0.10$; p<0.001). Again, this result may have occurred owing to increased likelihood of imaging segments of the same axon when those segments are closer together (e.g. within the same ROI). We confirmed that artifactual movement within the imaging field was not driving distance-dependent changes in partial correlations by comparing partial correlations between non-activity-dependent mCherry axons across distance. No change in partial correlation was observed with increasing distance (*Figure 4P*; $r = -0.03$; p=0.35).

Our results indicate that there is a strong correlation and coherence between axon segments of cholinergic and noradrenergic axons within the cerebral cortex, that these correlations extend for long distances (at least 3.5–4 millimeters), and that the major contributor to these correlations is a common signal related to behavioral state (e.g. uninstructed movements and arousal). The residual, partial correlations that remain once this common signal is subtracted had a similar distribution to that of correlations between non-activity-dependent axons, indicating that these correlations may

not represent behaviorally relevant signals. Thus, while there is a broad range of correlations between modulatory axons, we were unable to confirm that the range of correlations not explained by the global signal is representative of a heterogeneous signaling mechanism.

Another way to reveal heterogeneity of activity between simultaneously monitored ACh and NA axonal segments is to compare their activity for overall average axonal responses that are of similar magnitude (see *Figure 2—figure supplement 1K–O*). Indeed, even though the average response of the ACh of NA axons may be similar for two different bouts of whisker movement, the pattern of activity generated within the simultaneously monitored ACh or NA axon segment can vary strongly. These variations gave rise to the strong appearance of whisker bout to bout heterogeneity in axonal response patterns (see *Figure 2—figure supplement 1K–O*).

Recently, it has been reported that cholinergic receptor activation across the dorsal cortex is decorrelated in association with increases in movement, suggesting that the cholinergic fibers projecting to different regions of the cortex may decorrelate with increased behavioral engagement (*Lohani et al., 2022*). However, our results indicate that the ability to make this conclusion may be severely affected by the limited ability to monitor noradrenergic/cholinergic activity using fluorescent reporting methods. Indeed, we observed a striking decrease in correlation between cholinergic (and noradrenergic) axon segments during walking (*Figure 4—figure supplement 1A and B*). However, we obtained the same result when comparing adjacent segments from the same axon (which should remain highly correlated throughout all behavioral states), indicating that this movement-related decorrelation in fluorescence-based activity is most likely an artifact of frequency-limited fluorescence-based data collection methods. Since we cannot resolve individual action potentials (or the resulting fluctuations in $[Ca^{2+}]_i$) within axons at all frequencies, we are unable to measure whether there is, or is not, a decorrelation of biologically relevant activity between these axons during walking or whisking.

## Relationship between VCIN activity and behavioral state

In addition to input from the basal forebrain, the cerebral cortex receives local cholinergic signaling through local VIP+/ChAT+ interneurons (VCINs). To determine whether activity in this population is modulated by arousal and behavioral state, we recorded GCaMP6s activity from the somata of ChAT-expressing cortical interneurons across the dorsal cortex while monitoring arousal related behavioral variables (whisker movements, walking; example shown in *Figure 5A and B*). Cell bodies were selected for imaging from across the dorsal surface of the neocortex approximately 200 μm below the pial surface (*Figure 5C*), consistent with VCINs being most densely present in cortical layers 2/3 (*Dudai et al., 2021*).

To determine whether VCIN activity tracked behavioral state transitions, we aligned GCaMP6s fluorescence in VCINs to the onsets and offsets of whisker movements (twitches or larger whisking bouts), and walking bouts (*Figure 5D–F*, respectively). Averaged activity across all recorded VCINs tracked changes in the animal's behavioral state. Small whisker pad twitches were associated with a modest average increase in VCIN population activity, while longer whisking bouts and walking bouts were associated with larger increases in average population activity (*Figure 5D–F*). In similarity with both BF-ACh and LC-NA axons, VCIN activity quickly increased at the onset of facial movements and returned to baseline within 1–2 s following a return to stillness (*Figure 5E*). Examining the relationship between whisking onset and VCIN GCaMP6s activity at an expanded time base (*Figure 5E*, inset) revealed a slight (<150 ms) onset lag between whisking onset and VCIN GCaMP6s activity. VCIN activation was less tightly coupled to the onset and offsets of walking bouts (*Figure 5F*), likely due to facial movements consistently occurring prior to walking bout onsets and persisting after walking offset (*Figure 2—figure supplement 3A and B*). The observed relationship between VCIN signaling and whisking and walking behaviors was prominent in only a subset of these neurons (*Figure 5G–I*). To quantify the reliability of responsiveness of individual VCINs to behavioral events, we calculated the probability that dF/F of a given VCIN was significantly increased 1 s following event onset, as compared to 1 s prior to event onset (Student's *t*-test, alpha = 0.01; *Figure 5J*). On any given whisking bout 43.0 ± 31.3% of VCINs responded significantly while on any given walking bout 27.2 ± 27.9% responded significantly with this time period comparison.

To determine the extent to which VCIN signaling is uniform across cortical regions, we compared the correlation in activity across simultaneously imaged subpopulations (n = 4–15 cells per session; n = 178 total pairs) separated by a distance of up to 3.75 mm (*Figure 5K*). Correlation of activity

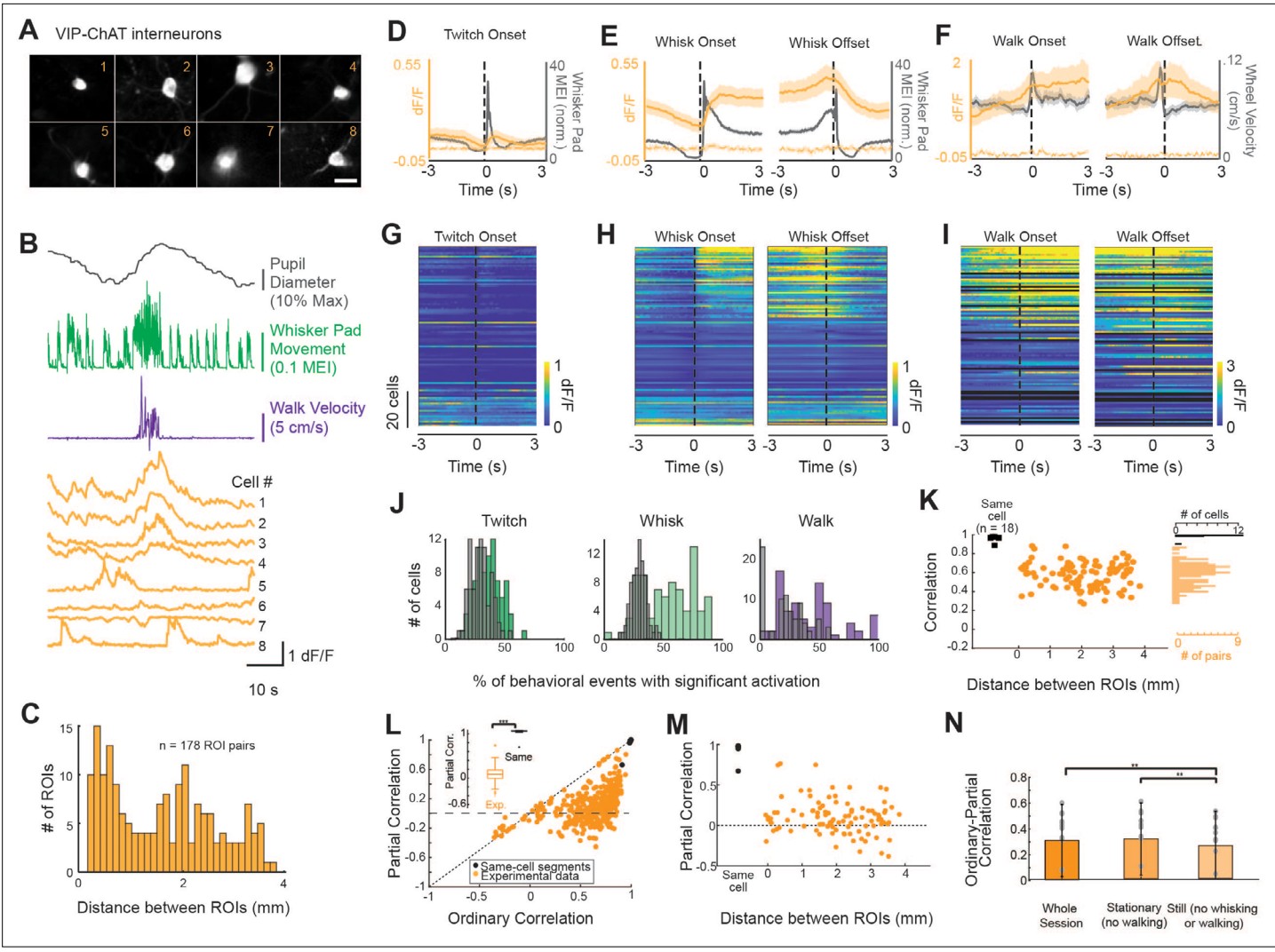

**Figure 5.** A subpopulation of VIP-ChAT cortical interneurons track changes in behavioral state.
(A) Images of eight simultaneously recorded VIP-ChAT cortical interneurons (VCINs). (B) Example recording taken from the eight somas shown in panel (A). Note the clear heterogeneity of VCIN response (yellow) during walking (purple) and whisking (green) activity. (C). Distribution of distance between regions of interest (ROIs) containing VCINs. ROIs were separated from 0.075 to 3.9 mm (N = 7 mice, n = 98 cells). (D–F) Group-averaged VCIN activity aligned to 870 twitching onsets, 1104 whisking onsets, 1015 whisking offsets, and 127 walking onsets and offsets (N = 6 mice, n = 84 cell bodies for walking data). Gray traces show the averaged change in whisker pad motion energy (D, E) or walking speed (F). Average +/- 95% confidence intervals are plotted. Dotted lines represent autofluorescent blebs, which showed no significant change in fluorescence with movement. Inset in (E) is at an expanded time base with normalization of data from base to peak for comparison of the kinetics of onset of whisker movements and VCIN activity. (G–I) Same data as displayed in panels (C–E) but separated to visualize heterogeneity of individual cell responses to behavioral state changes. Cells are sorted based on the magnitude of dF/F change at the onset of whisking. (J) The consistency of a single VCIN's response to behavioral state changes was quantified by calculating the percent of twitching, whisking, or walking bouts that were associated with significant increases (1 s post-onset versus 1 s pre-onset) in dF/F and compared to distributions attained with shuffled data. (K) Correlation between VCIN cells by distance between the ROIs. Measuring GCaMP6s from two halves of the same cell body resulted in an average correlation that was near 1 (black dots; N = 3 mice, n = 18 same-cell pairs). (L) Partial residual correlation (following removal of correlation with common signal) versus ordinary correlation reveals that a significant component of the ordinary correlation results from the common signal. Common signal was calculated as the average activity of the imaged neurons, excluding the two cells being correlated. ***p<0.001; Student's t-test. (M) Partial residual correlation as a function of the distance between ROIs containing the cells being correlated. The partial correlation decreases slightly with distance (r = –0.09; n = 98 cells; p=0.07). (N) Effect of restricting data to stationary or still periods on partial correlations. p<0.01; One-way ANOVA with multiple comparisons post-hoc testing.

between pairs of VCIN neurons varied from 0.43 to 0.96 and averaged 0.74 ± 0.12. As a control, we compared the correlation between GCaMP6s signaling in two halves of the same VCIN neurons (*Figure 5K*, black dots). We reasoned that any correlation below 1 would reveal limitations within our data set. Indeed, all (n = 18) of the adjacent cell body halves exhibited a correlation at, or near,

1 (minimum correlation = 0.91; *Figure 5K*). These correlations are higher than those we obtained by comparing two segments of the same axon (e.g. *Figures 4J and 5K*). We assume this difference is the result of the high dF/F signals, and larger imaging volumes, exhibited by cell bodies in comparison to axon segments. There appeared to be no significant relationship between distance and correlation between VCIN GCaMP6s activity (*Figure 5K*).

To examine the possibility that correlations between VCIN neuronal GCaMP6s signal was the result of co-correlation with a common variable, related to behavioral state, we performed partial correlations in which the common signal was the average VCIN activity of simultaneously imaged neurons (excluding the two cells being correlated; *Figure 5L*). For neurons with ordinary correlations less than approximately 0.6, the residual partial correlation centered around approximately 0, suggesting that these neurons were correlated in their activity owing to co-correlation with a common signal. Closer examination of the subset of pairs of neurons that exhibited ordinary correlations greater than approximately 0.6 revealed a population of neuronal pairs whose partial correlation was unusually positive, increasing toward a partial residual correlation of 0.84, as the ordinary correlation approached 0.96 (*Figure 5L*). This result suggests that this small number of neuronal pairs were strongly correlated in activity outside that associated with the common (average) VCIN activity.

Calculating partial correlation between two halves of the same cell body, taking into account the average common activity of other simultaneously imaged neurons, did not reduce the correlation in VCINs (ordinary correlation: 0.99 ± 0.02, partial correlation: 0.99 ± 0.03; t = –1.35, p=0.20), an expected result since these two cell body halves should be highly correlated in all of their activity, including that remaining following removal of the group common signal correlation. Plotting the partial residual correlation between VCIN neurons as a function of distance revealed a small decrease with distance (*Figure 5M*; *r* = –0.14; p<0.01), suggesting that nearby VCIN neurons are slightly more likely to be correlated outside the common signal than distant cells. Interestingly, in contrast to ACh axons, the reduction in correlation between VCIN neurons by taking into account a common signal was not dependent upon whisking or walking (*Figure 5N*).

## Discussion

In this study, we used GCaMP6s calcium imaging in cholinergic and noradrenergic cortical axons to determine the extent to which each neuromodulatory pathway may provide a global state-dependent signal to cortical regions, and whether or not other, more heterogeneous, activity may also be present. First, our results confirmed that BF-ACh and LC-NA activity across the cortex is strongly linked to changes in behavioral state, as indicated by externally observable measures (whisker/facial movements, pupil diameter, locomotion) with strong ACh/NA activations occurring during periods of behavioral engagement (*Figures 1–4*, *Figure 2—figure supplement 1*). Second, we found that facial/whisker movement is a better predictor of BF-ACh and LC-NA activity than walking speed or pupil diameter (*Figure 3*). Third, we found strong evidence that ACh and NA axonal signaling is dominated by a global, arousal/movement-related signal (*Figure 4*). Fourth, we found evidence consistent with a weaker heterogeneous component of ACh and NA axonal signaling (*Figure 4*, *Figure 2—figure supplement 1K–O*). Finally, we presented evidence for both behavior-linked and heterogeneous cortical ACh signaling via local cortical VIP-ChAT interneurons (*Figure 5*).

### Cholinergic and noradrenergic signaling related to arousal and behavioral engagement

BF-ACh and LC-NA have long been implicated in initiating and maintaining the waking state and arousal (*Berridge, 2008*; *McCormick, 1989*; *Picciotto et al., 2012*). Early work exploring the role of the noradrenergic neurons of the LC strongly suggested that this region is important for triggering sleep–wake transitions and variations in behavioral engagement and vigilance. For example, activity of LC neurons is higher during waking compared to sleeping (*Hobson et al., 1975*), and increased LC activity occurs before the transition from sleep to awake (*Aston-Jones and Bloom, 1981*). The potential causal role of the LC in regulation of wakefulness is supported by more recent experiments that have shown that optogenetic stimulation of the LC induces a sleep–wake transition (*Carter et al., 2010*; *Hayat et al., 2020*). Noradrenergic signaling is also associated with moment-to-moment fluctuations in arousal and behavioral engagement/vigilance within the waking state (*Joshi et al., 2016*; *Joshi and*

*Gold, 2022*); reviewed in (*Aston-Jones and Cohen, 2005*; *Poe et al., 2020*). In the mouse, activity of noradrenergic axons in primary visual and auditory cortices tracks fluctuations in pupil diameter and orofacial movements (*Larsen and Waters, 2018*; *Reimer et al., 2016*), which, in mice, are indicators of behavioral state (reviewed in *McGinley et al., 2015b*). Moreover, optogenetic activation of the LC induces pupil dilation while inhibition of the LC leads to pupil constriction (*Breton-Provencher and Sur, 2019*; *Megemont et al., 2022*; *Reimer et al., 2016*). Early recordings from the LC in awake, behaving rodents or monkeys revealed tonic increases in average firing rate related to arousal and phasic, often synchronous, discharge associated with short-term increases in behavioral vigilance to external environment and stimuli (reviewed in *Aston-Jones and Cohen, 2005*; *Sara, 2009*; *Sara and Bouret, 2012*). More recent investigations reveal at least some degree of heterogeneity and specialization in LC neuronal signaling (see below; *Breton-Provencher et al., 2022*; *Chandler et al., 2014*; *Poe et al., 2020*; *Totah et al., 2019*; *Uematsu et al., 2017*). Our observation of a common signal across noradrenergic axons related to behavioral state (e.g. whisker movements – a general measure of behavioral engagement in mice) is consistent with a global noradrenergic influence on cortical and subcortical network activity related to behavioral arousal, engagement, and vigilance.

Similar to the LC, the basal forebrain cholinergic system has been associated with transition to the waking state from sleep (*Han et al., 2014*; *Irmak and de Lecea, 2014*; *Teles-Grilo Ruivo et al., 2017*; *Xu et al., 2015*), as well as general cortical activation (*Buzsaki et al., 1988*; *Metherate et al., 1992*; *Metherate and Ashe, 1993*). During slow wave sleep, ACh release in the cortex and thalamus is substantially reduced compared to the waking state and REM sleep state (*Lee et al., 2005*; *Williams et al., 1994*) and stimulation of the cholinergic basal forebrain promotes wakefulness (*Han et al., 2014*; *Xu et al., 2015*; *Zant et al., 2016*). In the waking state, cortical cholinergic axonal activity fluctuates in relation to increases in behavioral arousal and engagement (as measured by the onset of whisker movements, increases in pupil diameter and locomotion) (*Eggermann et al., 2014*; *Nelson and Mooney, 2016*; *Reimer et al., 2016*). Activation of the basal forebrain cholinergic system appears to be particularly important for the suppression of spontaneous slow oscillations in cortical and thalamocortical activity, which are characteristic of behavioral quiescence or drowsiness (*Eggermann et al., 2014*). Our observation of a global common signal in cholinergic axons correlated with measures of behavioral engagement, particularly movements of the whiskers and face, supports the view that one of the major roles of the basal forebrain cholinergic system is to provide a broad modulation of the neocortex in association with moment-to-moment changes in behavioral state, including arousal, engagement, vigilance, and attention.

Interestingly, a recent study of dopaminergic axons within the mouse cortex also revealed state-dependent activity very similar to that observed here for cholinergic and noradrenergic axons, including increases in activity with whisker movement, pupil dilation, and locomotion (Figure 4 in *Sturgill et al., 2020*). These results, along with studies of multiple cell types throughout the cortex, thalamus, and other structures (*McCormick et al., 2020*; *McGinley et al., 2015b*; *Musall et al., 2019*; *Nestvogel and McCormick, 2022*; *Steinmetz et al., 2019*; *Stringer et al., 2019*), reveal frequent coordinated rapid shifts in neural dynamics throughout the mouse brain during the waking state that are highly coordinated with rapid changes in behavioral state. This coordinated shift in neural dynamics across broad regions of the brain, including neuromodulatory pathways, does not preclude important regional and temporal variations in activity that are specific to each pathway/system. For example, we have shown here that the activation of the basal forebrain cholinergic pathways remains sustained throughout prolonged bouts of movement, whereas the noradrenergic system from the locus coeruleus is more transient – mainly signaling the onset of a period of behavioral engagement (*Figure 2D and E*). This result suggests that the cholinergic system is more associated with alteration in brain dynamics associated with sustained engagement with the external world, while the noradrenergic input may be more related to the behavioral transition to external engagement (*Aston-Jones and Cohen, 2005*; *Clayton et al., 2004*; *Poe et al., 2020*; *Reimer et al., 2016*; *Vazey et al., 2018*). Revealing the unique spatiotemporal features of each neuromodulatory system will be critical to understanding how each contributes to the control of neural dynamics in the waking state.

## Homogeneous and heterogeneous ACh and LC-NA signaling

In addition to a global behavioral state-linked signal, ACh and LC-NA may also provide heterogeneous signals throughout the brain. These targeted signals may support a range of brain processes and

behaviors, including selective sensory processing, cortical plasticity, and memory formation (*Aston-Jones and Cohen, 2005*; *Breton-Provencher et al., 2022*; *Deitcher et al., 2019*; *Ego-Stengel et al., 2001*; *Goard and Dan, 2009*; *Janitzky et al., 2015*; *Nelson and Mooney, 2016*; *Pinto et al., 2013*; *Polack et al., 2013*; *Sara, 2009*; *Sara and Bouret, 2012*; *Uematsu et al., 2017*; *Usher et al., 1999*).

Specificity of cholinergic cortical signaling has been demonstrated both anatomically and functionally. Inputs and outputs of cholinergic neurons in the BF are topographically organized (*Gielow and Zaborszky, 2017*; *Zaborszky et al., 2015*) and activating distinct regions of the BF results in desynchronization of restricted cortical regions (*Kim et al., 2016*). Recent work has demonstrated heterogeneity in cortical ACh receptor activation, particularly during periods of locomotion (*Howe et al., 2019*; *Lohani et al., 2022*), and ACh receptor subtype distribution is highly non-uniform across the brain (*Picciotto et al., 2012*). Simultaneous monitoring of activity of cholinergic neurons in different parts of the basal forebrain system during the learning and performance of a variable reinforcement task in mice reveal both broadly coordinated activity as well as consistent differences in the patterns of cholinergic neuronal activity in relation to their projection targets (*Robert et al., 2021*).

The extent to which the LC-NA system is modular, or provides targeted neuromodulation throughout the brain, remains under debate, although evidence for modularity is accumulating (*Breton-Provencher et al., 2022*; *Chandler et al., 2019*; *Poe et al., 2020*; *Totah et al., 2018*; *Totah et al., 2019*; *Uematsu et al., 2017*). While some anatomical and functional evidence supports the view that neurons within the LC-NA system share a broadly similar connectivity and functional effect, and that the activity of LC neurons are often synchronized (*Deitcher et al., 2019*; *Kim et al., 2016*; *Schwarz et al., 2015*), more recent investigations have emphasized the presence of genetic, anatomical, and functional specializations within sub-regions of the locus coeruleus (reviewed in *Poe et al., 2020*; *Totah et al., 2018*). Indeed, two recent reports on the activity and influence of subregions of the LC in mice performing learned tasks provide evidence for both broad NA coordination, as well as functional specialization between different neuronal groups within the LC (*Breton-Provencher et al., 2022*; *Uematsu et al., 2017*).

Even if the action potential activity of modulatory axons was uniform across the neocortex, it is still possible that the release and actions of these neurotransmitters are locally modulated. For example, presynaptic modulation of noradrenergic and cholinergic release or modulation of postsynaptic receptor–effector mechanisms can be locally specific (*Lovinger et al., 2022*; *Pardi et al., 2020*; *Sarter et al., 2014*), indicating that our data does not distinguish between heterogeneity owing to local modulation from variations in action potential activity across the active population of cholinergic or noradrenergic neurons. Another important caveat to keep in mind is that our fluorescence monitoring of noradrenergic and cholinergic axonal activity is limited in frequencies below approximately 1–2 Hz. Therefore, we are unable to determine if activity in these axonal pathways are widely synchronized or desynchronized at frequencies above this low frequency cutoff.

Our data is consistent with the LC-NA and BF-ACh systems broadcasting both a broad low-frequency state-dependent signal over wide regions of the neocortex, as well as more heterogeneous activity that may be regionally or temporally separate and distinct. Evidence supporting the presence of a broadly coordinated signal is the high degree of correlation between the GCaMP6s activity in axonal segments that could be explained by a common signal (*Figure 4*). Evidence for a more heterogeneous signal is provided by the large range of activity that remains once the common signal is removed and the trial-to-trial variation in axonal segment activity, even on trials in which activation of that modulatory system was nearly identical (*Figure 2—figure supplement 1K–O*). In other words, the broad population of axon segments do not behave as if they are all carrying the same activity, which is the case when comparing segments from the same axon (*Figure 4*).

## ACh signaling from cortical interneurons

ACh signaling is complicated by the presence of two distinct cholinergic systems that modulate activity in the cortex: the well-characterized long-range projections from the BF discussed above and a population of cortical interneurons (VCINs) that are equipped to modulate local cortical circuits through both GABAergic and cholinergic signaling. VCINs are functionally and spatially situated as a prime candidate to enhance regionally specific modulatory signaling of the cholinergic system. VCINs have been reported to be sparsely distributed throughout the cortex, primarily in layers 2/3 (*Dudai et al., 2021*), and they possess a combination of GABAergic and cholinergic axon terminals, arranged

in a highly target-specific manner, ideal for circuit-specific modulation. Individual axon segments of VCINs can contain both VAChT+ and VAChT- terminals, suggesting specificity based on postsynaptic targets (*Dudai et al., 2021*; *Granger et al., 2020*; *Obermayer et al., 2019*).

The present work provides evidence that, like the long-range cholinergic BF projections to cortex, some local VCINs track the arousal and behavioral state of the animal, exhibiting increased activity related to behavioral state changes (*Figure 5D–F*). However, amongst the population of VCINs, there is clear heterogeneity in the cells' responsiveness to behavioral state, with only a subset of VCINs across the cortex displaying activity that strongly and reliably tracks behavioral state variables (*Figure 5B, G–I*). This heterogeneity of activity within local cholinergic signaling may reflect circuit-specific arousal-related modulation of cortical dynamics, or it may be evidence that there is a functionally distinct subgroup of VCINs, which are less strongly influenced by behavioral/arousal state changes. The presence of subsets of VCIN interneurons that exhibit broadly synchronized activity, even when these neurons are distant from one another (*Figure 5M*), suggests the presence of a common synchronizing input. Identifying the mechanisms of this synchronization, particularly in relation to variations in behavior, will be important to understanding the role of these interneurons in cortical dynamics.

## Conclusions

Together, our findings provide evidence that a significant portion of activity within cholinergic and noradrenergic cortical projections is tightly coupled to spontaneous behavioral state changes. By simultaneously imaging axonal activity across functionally distinct cortical areas separated by up to ~4 mm, we demonstrate that both BF-ACh and LC-NA projections provide a global, arousal/behavioral state-related signal to the neocortex. However, our data is also consistent with heterogeneous signaling being provided by local cortical cholinergic interneuron activity, as well as activity in BF-ACh and LC-NA axons.

# Materials and methods

## Animals

All experiments were approved by the University of Oregon Institutional Animal Care and Use Committee. Experiments were conducted using male and female mice aged between 6 and 10 wk at study onset. All mouse strains used in this study were of C57BL/6J (IMSR Cat# JAX:000664, RRID:IMSR_JAX:000664) background and were purchased from Jackson Laboratory and bred in-house. ChAT-cre mice (IMSR Cat# JAX:006410, RRID:IMSR_JAX:006410), which express cre-recombinase in cholinergic neurons, were used for viral injection experiments to record from BF-ACh axonal projections. Viral injections were used for BF-ACh studies to avoid imaging axons or dendrites from cholinergic projections not arising from the BF (e.g. cortical cholinergic interneurons). For LC-NA experiments, DBH-cre mice (IMSR Cat# JAX:033951, RRID:IMSR_JAX:033951), which express cre-recombinase in noradrenergic neurons, were crossed with Ai162 (IMSR Cat# JAX:031562, RRID:IMSR_JAX:031562) mice, which cre-dependently express GCaMP6s. ChAT-cre mice crossed with Ai162 mice were used for VCIN studies since these mice express GCaMP6s in cholinergic cells and projections, including VCINs. All mice were individually housed under an inverted 12:12 hr light/dark regime and had access to food and water ad libitum. All experiments were conducted during the active dark cycle in reverse light-cycled mice.

## Surgical procedures

All surgical procedures were performed in an aseptic environment under 1–2% isoflurane anesthesia (oxygen flow rate: ~1.5 L/min) and mice were homeothermically maintained at 37.5°C under systemic analgesia (Meloxicam SR: 6 mg/kg, s.c.; Buprenorphine SR: 0.05 mg/kg, s.c.). For viral injection experiments, two small craniotomies (~1 mm) were made for bilateral viral injections into the basal forebrain (from bregma: 1.44 mm lateral, 0.6 mm posterior). The pipette was lowered 3.8 mm ventral to the surface of the brain and approximately 1 µl of an axon-targeted GCaMP6s adeno-associated virus (pAAV-hSynapsin1-FLEx-axon-GCaMP6s, addgene; *Broussard et al., 2018*) solution was injected. Animals were allowed to recover from the viral injection surgery before any subsequent surgeries. Imaging was performed at least 3 wk after virus injection to allow adequate time for axonal transport to the cortex.

For two-photon imaging, a titanium headpost was affixed to the skull with dental cement after removing the overlying skin and fascia. Then, an 8 mm circular craniotomy was made overlying the dorsal cortex using a dental drill. An 8 mm circular glass coverslip was placed in the craniotomy and the coverslip was affixed to the skull with flow-it composite (Flow-It ALC, Pentron) and a thin layer of dental cement. After all surgeries, mice recovered in an ~32°C recovery chamber and postoperative subcutaneous lactated ringer's solution was administered for 1–3 d as required.

## Experimental design

### Arousal measures

Previous studies of head-fixed mice have demonstrated rapid variations between distinct behavioral states, including sedentary (non-walking, non-whisking), whisker twitching, whisking, and walking (*McCormick et al., 2020*; *McGinley et al., 2015b*; *Musall et al., 2019*; *Salkoff et al., 2020*; *Stringer et al., 2019*). Since facial movements (particularly of the whiskers), pupil diameter, and locomotion have all been shown to be useful for quantifying behavioral state, we monitor these behavioral variables here. For all experiments, mice were head-fixed atop a cylindrical running wheel and video of the mouse face was acquired at 30 Hz using a Teledyne camera. Pupil diameter was measured both in real time and offline using a custom LabVIEW script (*McGinley et al., 2015a*; *Salkoff et al., 2020*). Running speed was measured using a rotary encoder (McMaster-Carr) attached to the wheel. Note that this study was not optimized for pupil diameter measures since we found it necessary to use low-background light levels for imaging fluorescent axons. Thus our coherence results between pupil diameter and cholinergic/noradrenergic axon activity may be lower than previously reported (*Reimer et al., 2016*). The running wheel was supported by springs and the rotary encoder also detected vertical movements of the running wheel, which can occur both with, and between, bouts of running. Whisker-pad movement was measured by selecting an ROI over the whisker pad and calculating the motion energy index (MEI) across the video. MEI was defined as the sum of the absolute change in pixel intensity within the ROI between adjacent video frames. Similarly, measures of movements of the jaw and snout were measured as MEI within a local region of the video, including these facial features (*Figure 3*). All waveform and trigger signals were digitized through a Micro1401 or a Power 1401, and collected using Spike2 version 7 or 8.

### Two-photon imaging

Two-photon imaging was conducted using a ThorLabs Multiphoton Mesoscope (Excitation NA 0.6, Collection NA 1.0) equipped with a 12 kHz Resonant Scanner and Virtually Conjugated Galvo Scanner Set along with a 1 mm range Remote Focusing Unit, allowing for rapid imaging across multiple ROIs varying in X, Y, and Z coordinates (*Sofroniew et al., 2016*). Excitation of GCaMP6s was achieved via a Ti-sapphire laser tuned to 920 nm (MaiTai, Spectra Physics). Scan Image software (Vidrio) was used for all imaging sessions. For cell body imaging, a minimum resolution of 0.5 μm/pixel was used, and for axonal imaging a minimum resolution of 1 μm/pixel was used to clearly delineate borders of somas or axonal processes. Fields of view ranged between 50 × 50 and 400 × 400 μm and were obtained at a minimum frequency of 10 Hz (range of approximately 10–30 Hz), based on known GCaMP6s kinetics (*Chen et al., 2013*).

Five mice were used in the BF-ACh study. Across 15 sessions, 397 axon segments were imaged. In the LC-NA study, six mice were used. Across 18 sessions, 283 axon segments were imaged. Seven mice were used for the VCIN study. Across 12 sessions, 98 cell bodies were imaged. Recording duration varied between ~10 and 40 min.

For motion control experiments, non-calcium-dependent mCherry was imaged alongside calcium-dependent GCaMP6s. Simultaneous excitation of mCherry and GCaMP6s was achieved via a Ti-sapphire laser tuned to 970 nm (MaiTai, Spectra Physics). Otherwise, all imaging procedures for the mCherry experiment were identical to those of GCaMP6s axon imaging studies. A total of 93 mCherry axons were recorded in three mice for this study.

### Histology

Mice were euthanized with 51% carbon dioxide (24 L/min) and transcardially perfused with 25 mL of 0.01 M PBS followed by 20 mL of 4% paraformaldehyde. Brains were extracted and placed in 4%

paraformaldehyde for 24 hr at 4°C and subsequently transferred into 20% and 30% sucrose for 24 hr at 4°C. Brains were then sliced into 55 µm sections at –21°C using a cryostat (Leica).

Antibody staining was utilized to identify the expression of antigens of interest, as well as enhance GCaMP6s expression (*Daigle et al., 2018*; *Larsen et al., 2018*). We performed an antigen retrieval procedure where sections were incubated in 10 mM sodium citrate (pH 6.0) and 0.05% Triton-X-100 for 20 min at 75°C. Sections were then rinsed in 0.2% Triton-X-100 and 0.01 M PBS before being blocked with 5% normal donkey serum and 0.2% Triton-X-100 in 0.1 M PBS for 1–3 hr at room temperature. Sections were incubated for 48–72 hr at 4°C in primary antibodies: anti-choline acetyltransferase 1:300 (Abcam Cat# ab178850, RRID:AB_2721842), anti-tyrosine hydroxylase 1:300 (Abcam Cat# ab137869, RRID:AB_2801410), and/or anti-GFP 1:2000 (Abcam Cat# ab6673, RRID:AB_305643). Primary antibodies were diluted in the blocking buffer solution. Tissue was then rinsed in 0.2% Triton-X-100 and 0.01 M PBS before being incubated for 1 hr at room temperature in Alexa 488 (Abcam Cat# ab150129, RRID:AB_2687506) and 555 (Abcam Cat# ab150074, RRID:AB_2636997), conjugated secondary antibodies (1:500) in 0.1 M PBS. After the secondary antibody incubation, sections were rinsed in 0.2% Triton-X-100 and 0.01 M PBS followed by 0.01 M PBS before being mounted on glass slides and cover-slipped with prolong gold antifade mounting media. Images are displayed as mean intensity z-projections and were captured with a Nikon SoRa spinning disk confocal microscope using 20 X/0.75 NA and 40 X/1.15 NA objectives. Cholinergic axons in cortex (*Figure 1—figure supplement 1A and B*) were captured at ×160 by adding a ×4 additional objective to the light path while imaging with the ×40/1.15 NA objective. GCaMP6s and neuromodulatory marker expression in cell bodies were quantified by hand and used to determine the extent of colabeling (see *Figure 1—figure supplement 1*).

## Statistical analysis
### Behavioral data analysis
All imaging data was first aligned to behavioral data using custom MATLAB scripts. Pupil diameter was normalized to the max pupil diameter in each session to limit overestimation of arousal state in low lighting conditions. FaceMap (*Syeda et al., 2022*; https://github.com/MouseLand/FaceMap) was used to calculate motion energy of the snout and mouth. Onsets and offsets of walking bouts, whisking bouts, and facial twitches were detected using custom MATLAB code. Walking bouts were defined as times during which the mouse exceeded a speed of 2.5 cm/s for at least 2 s. Whisker pad motion energy was normalized between 0 and 1 for each session, and whisking bouts were defined as periods in which the normalized whisker pad motion energy exceeded 20% for more than 1 s. Twitches were defined as moments when the normalized whisker pad motion energy exceeded 20% for less than 0.5 s. Walking bouts, whisking bouts, and twitches that were not preceded and followed by 1 s of stillness were excluded. This requirement led to a drop in axonal fluorescence prior to the onset of movement (e.g. *Figure 2A and B*). For the BF-ACh study, GCaMP6s activity in 397 axon segments was compared to 101 waking onsets and offsets, 330 whisking onsets, 201 whisking offsets, and 1108 twitch onsets. For the LC-NA study, GCaMP6s activity in 283 axon segments was compared to 34 walking onsets and offsets, 400 whisk onsets, 353 whisk offsets, and 644 twitch onsets. For the VCIN study, GCaMP6s activity in 84 cell bodies was compared to 127 walking onsets and offsets and activity in 98 cell bodies was compared to 1104 whisking onsets, 1015 whisking offsets, and 870 twitch onsets. For the mCherry motion control study, fluorescence in 93 axons was compared to 622 walking onsets and offsets, 3342 whisking onsets, 3137 whisking offsets, and 1129 twitch onsets.

### Two-photon imaging analysis
Suite2P (*Pachitariu et al., 2017*) was used to identify cells and axons from two-photon mesoscope imaging data. Axon segments and cell bodies were imaged in several (up to four) ROIs chosen across the dorsal cortex. Distances between ROIs (*Figures 1, 4 and 5*) were determined based on the center point of each ROI. ROI size ranged from 50 × 50 to 400 × 400 µm.

All identified cells and axonal shafts were manually verified. A Suite2P-defined region was confirmed to be an axon if all pixels identified were within ~5 µm of the axon and presynaptic boutons and if a significant (>~20 µm) length of the axonal segment was within the region (e.g. see *Figure 4A*). A Suite2P-defined cell was considered to be a cell only if nearly all identified pixels were within a cell body. Any identified regions that did not correspond to a clearly identifiable distinct cell body or axonal process were excluded. Axons and cell body regions that were identified with Suite2P but were

visually confirmed (i.e. clear continuity of axon form, within the imaging plane, connecting the two segments together) to belong to the same cell or axon were removed from our experimental data set and included in our same-axon/cell data set if they were of a similar size and shape to those selected in the experimental data set (ACh: n = 18; NA: n = 30; VCIN: n = 18).

To control for the possibility of motion artifacts, autofluorescent 'blebs' that did not have calcium-dependent activity and were approximately of similar size to axonal varicosities were selected from within the imaging fields (ACh: n = 22; NA: n = 19; VCIN: n = 8). Since these autofluorescent blebs are not necessarily a similar thickness or brightness to our experimental data, it is possible that movement in the Z plane is not entirely accounted for by comparing our experimental data to blebs. Therefore, we performed an additional control experiment in which we injected a cre-dependent mCherry virus (pAACV-hSyn-DIO-mCherry; a gift from Bryan Roth; Addgene plasmid #50459; http://n2t.net/addgene:50459; RRID:Addgene_50459) into the basal forebrain of three ChAT-cre mice. We then recorded fluorescence from 93 axons in 30 ROIs (between 5 and 8 ROIs recorded simultaneously each session). This control allowed us to determine whether there was any effect of movement of axons in and out of the imaging plane that could contribute to our observed experimental results. Preprocessing and analysis of mCherry control axons were performed in the same manner as for experimental data (described below).

Preprocessing of calcium fluorescence data was performed using methods previously described (*Nelson and Mooney, 2016*; *Reimer et al., 2016*). Briefly, all traces were upsampled to 100 Hz and low-pass filtered to 10 Hz. Signal-to-noise ratios were then calculated for all traces by dividing the max power between the frequency range of 0.05–0.5 Hz by the mean power between 1 and 3 Hz. All traces that did not meet a minimum signal-to-noise criteria of log(20) were excluded from further analyses. Normalized fluorescence values were used in statistical analyses to avoid overweighting larger or brighter axon segments. For displaying data, ΔF/F was calculated for each identified axon or cell body using the median fluorescence value over each recording session as the baseline fluorescence. All preprocessing was performed using custom MATLAB scripts.

## Relationships between behavior and axon activity

To determine the frequencies at which axonal GCaMP6s fluorescence and arousal-related behaviors (walking, pupil dilation, facial movement) were most correlated, magnitude-squared coherence was calculated using a 2 min window with a 98% overlap and a Hamming filter using the MATLAB function mscohere. As a control, axon activity traces were shuffled in time and coherence was calculated with all behavioral variables.

To assess the timing of axonal activity (GCaMP6s fluorescence) in comparison to arousal-related behaviors, cross-correlations between axon activity and behavioral measures were calculated using the MATLAB function xcorr with a 3 s maximum lag. Cross-correlations were calculated after low-pass filtering at 1 Hz to capture the frequencies at which our behavioral variables and GCaMP6s have the most power (*Figure 2—figure supplement 2*).

Ridge regressions were used to determine the relative strength of the relationships between behavioral variables and axon activity. All data was first low-pass filtered at 1 Hz. Models were cross-validated to account for overfitting.

## Relationships between axons/cells

Magnitude-squared coherence and cross-correlation was calculated between all pairs of simultaneously recorded axon segments using the same parameters used to relate axon activity to behavior, as described above. Distance between axons was estimated by calculating the distance between the center point of each ROI. Therefore, axons within the same ROI were coded as 0 mm apart.

Since our data may be strongly influenced by the frequency of activity occurring in the axons (owing to the preference of GCaMP6s fluorescence for low frequencies: *Figure 2—figure supplement 2*), we sought to better understand the limitations of our activity monitoring methodology by comparing the coherence and correlation of activity in two adjacent segments of the same axon (*Figure 4*, *Figure 2—figure supplement 2*). Since axon conduction failures are rare (*Foust et al., 2010*; *Popovic et al., 2011*), we reasoned that any decrease below 100% coherence or below a correlation of 1 between activity in adjacent segments of the same axon most likely indicates a fundamental limitation

of fluorescence-based data. This limitation results from the complex relationship between axonal action potential activity, $[Ca^{2+}]_i$, GCaMP6s, and our ability to monitor axonal GCaMP6s fluorescence.

To this end, first we identified several (n = 18 ACh; n = 30 NA) pairs of adjacent axon segments that we determined by eye to be portions of the same axon (i.e. segments joined by a clear continuity of axonal form). Plots of coherence between adjacent segments of the same axon revealed high (e.g. ≥0.8) values at frequencies below approximately 1 Hz, but this coherence decreased strongly at frequencies above approximately 1 Hz, such that the coherence between the axon segments was similar to that obtained with shuffled data at frequencies >approximately 3 Hz (*Figure 4D and E*). This result indicates that our measure of axonal activity is limited to variations in action potential firing rate below approximately 1 Hz. Therefore, we applied a 1 Hz low-pass filter to the recordings prior to calculation of correlation. Examining different cutoff frequencies for the low-pass filter indicated that a cutoff frequency of 1 Hz was optimal for enhancing correlation in same-axon data, while preserving information (see *Figure 4—figure supplement 1D and F*).

Since we observed a broad distribution in cross-correlations between simultaneously recorded axons or cells (*Figure 4J, K and M*), we used a partial correlation analysis to determine whether some pairs were highly correlated due to a shared 'common signal.' Partial correlations were calculated by correlating the residuals of a linear regression between a controlling variable, or common signal, and each of two axons or cells in a pair using the MATLAB function partialcorr. The average dF/F of all simultaneously recorded axons or cells (excluding the two axons or cells being compared) was used as a controlling variable. Counts of simultaneously imaged axons or cells ranged from 7 to 61 for ACh, 9 to24 for NA, and 3 to 10 for VCINs. Since our data suggest that facial activity could contribute to the common signal, we also used whisking activity as the controlling variable in a separate analysis (*Figure 4—figure supplement 1H,I*). Partial correlations were calculated after low-pass filtering at 1 Hz. To determine the impact of behavioral state on partial correlation, periods in which the animal was stationary (not walking) or still (neither walking nor whisking) were isolated (e.g. *Figure 4—figure supplement 1J*).

Custom MATLAB scripts used in data analysis are available online at https://www.github.com/lncollins91/ACh_NA_VCIN, (*Collins, 2023* copy archived at swh:1:rev:6c03e912e115a52cf9118a777fa9aabfa6f17507).

## Acknowledgements

This work was funded by the National Institutes of Health (NIH) R35NS097287 and R01NS118461. We would like to acknowledge all members of the McCormick lab for helpful discussions while preparing this manuscript and Evan Vickers for building critical components of the imaging system used for collection of the data presented here.

## Additional information

### Funding

| Funder | Grant reference number | Author |
| --- | --- | --- |
| National Institutes of Health | R35NS097287 | David A McCormick |
| National Institutes of Health | R01NS118461 | David A McCormick |

The funders had no role in study design, data collection and interpretation, or the decision to submit the work for publication.

### Author contributions

Lindsay Collins, Formal analysis, Investigation, Writing – original draft, Writing – review and editing; John Francis, Brett Emanuel, Investigation; David A McCormick, Conceptualization, Funding acquisition, Writing – original draft, Writing – review and editing

## Author ORCIDs

Lindsay Collins (ID) http://orcid.org/0000-0002-8588-2780
Brett Emanuel (ID) http://orcid.org/0009-0001-8494-728X
David A McCormick (ID) http://orcid.org/0000-0002-9803-8335

## Ethics

All experiments were approved by the University of Oregon Institutional Animal Care and Use Committee and performed in strict accordance with the recommendations in the Guide for the Care and Use of Laboratory Animals of the National Institutes of Health. All surgery was performed under isoflurane anesthesia, and every effort was made to minimize suffering.

## Decision letter and Author response

Decision letter https://doi.org/10.7554/eLife.81826.sa1
Author response https://doi.org/10.7554/eLife.81826.sa2

---

# Additional files

## Supplementary files

• Transparent reporting form

## Data availability

Data files have been deposited to the Open Science Framework (https://osf.io/rwtpu/). Custom Matlab codes can be found at https://www.github.com/lncollins91/ACh_NA_VCIN, (copy archived at swh:1:rev:6c03e912e115a52cf9118a777fa9aabfa6f17507).

The following dataset was generated:

| Author(s) | Year | Dataset title | Dataset URL | Database and Identifier |
|---|---|---|---|---|
| Collins L, Francis J, Emanuel B, McCormick DA | 2023 | Cholinergic and noradrenergic axon imaging | https://osf.io/rwtpu/ | Open Science Framework, rwtpu |

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
