## [Editor Report]

This study uses behavioral monitoring and cutting-edge calcium imaging approaches to track the activity of cholinergic and noradrenergic axons in the cortex of head-fixed mice, and correlate activity with behavioral state. The authors provide compelling evidence that behaviorally related signals are broadly broadcasted to the dorsal cortex and that there is also heterogeneity across axons and areas, independent of artifacts associated with these difficult measurements.

---

## [Decision Letter]

**Decision letter after peer review:**

Thank you for submitting your article "Cholinergic and noradrenergic axonal activity is both strongly coordinated with behavioral state and heterogeneous across the dorsal cortex" for consideration by *eLife*. Your article has been reviewed by 3 peer reviewers, and the evaluation has been overseen by Sacha Nelson as Reviewing Editor and John Huguenard as the Senior Editor. The following individual involved in review of your submission has agreed to reveal their identity: Michael London (Reviewer #3).

Essential revisions:

1) It will be essential to convince readers, including the editors and reviewers that the observed heterogeneity is not merely a reflection of motion artifacts. In discussion, it was recognized that this is difficult and might require new analyses or even new experiments, however it is a technical issue that goes to the heart of one of the main conclusions of the paper and so should be addressed rigorously.

2) Many details of the presentation were suboptimal and would benefit from revision. These include:

a) Improved clarity in the description of the methods, for example in the methods used to separate sources of variance in figure 3 and elsewhere.

b) Earlier and increased use of example recordings of single axon segment activity.

c) Improved reproduction of figure 5.

d) A discussion that more directly addresses the significance of the present findings.

*Reviewer #1 (Recommendations for the authors):*

1) As stated before, the manuscript needs better controls for movement-induced artifacts as they might contribute to the heterogeneity in the recorded signals across axons. A control in which motion artifacts are quantified without a biased selection criterion is needed to better support the claim of heterogeneity. Ideally- Ca+ insensitive structures of the same size as the axonal ROIs might be already present in the recordings and can be selected with an unbiased criterion. Selecting particles of the same size and shape as axonal varicosities exhibiting both red and green fluorescence is a possibility, as GCaMP-expressing axons should not be red fluorescent. Adding some control animals expressing GFP instead GCaMP to the dataset and repeating the different metrics is also a possibility.

2) While the prospect that the measured signals do not completely relate to action potential induced changes in axonal calcium is raised in lines 410-413, the manuscript does not explicitly address there or later in the discussion that the observed heterogeneity might arise from local, axon-specific presynaptic modulations and not because of differences in the spiking activity across axons (see for example Pardi et al. Science 2020). In the case the heterogeneity is not caused by movement-induced artifacts, these two alternative explanations should be explicitly addressed when interpreting the results.

3) Besides controlling for movement artifacts using an unbiased criterion for selecting calcium-insensitive structures, the manuscript would benefit with more extensive controls of the potential contribution of movement induced artifacts to the different metrics:

a. When measuring the probability that a given axon will respond to a behavioral event (lines 191-201). What are the values for non-GCaMP (bleb) , fluorescent particles ?

b. How does correlation between Ca+-insensitive structures depend on distance (like in Figure 4J).

4) While the examples in Figure 1 make very clear the existence of a global signals across regions and axons, none of the examples illustrates well the heterogeneity in the signals. Better exemplars showing the heterogeneity picked up by the analyses in Figure 4 would strengthen the claims.

5) Line 1006. What criteria is used to determine if an ROI corresponds to an axon? Are most ROIs on varicosities as stated in line 1012 when describing blebs, or on axonal shafts as in the examples on Figure 4A? If they are mostly on varicosities, analyses from same axon pairs should also come from ROIs of the same size and shape as the rest of the data, as the size and shape might influence correlation values.

6) What constitutes a 'bleb"? How are they distinguished from regular varicosities besides the lack of calcium-dependent activity? What is their size and shape compared to regular axonal ROIs?

7) Histological analysis of the GCaMP6 labelling is needed to control if the intended structures are the only ones labelled with GCaMP in both groups.

8) The original reference of the axon-GCaMP sensor used should be cited, not only the supplier of the virus (line 943).

9) Figure 3B does not have a time scale.

10) It is better to avoid using non metric units (line 952).

*Reviewer #2 (Recommendations for the authors):*

The authors miss the opportunity to compare behavioral state modulation across vs within cortical regions – that is, how similar or different is the "common signal" between different regions of cortex? They have these data (an advantage of the mesoscopic scale), and such an analysis would add significantly to the impact of the study.

The Results section is high in jargon and short on technical details related to the analyses shown, which are necessary for the reader to fully understand the meaning and potential significance of the analysis. This section would benefit from a clearer narrative about what exactly each analysis adds to the overall picture so the reader does not have to keep toggling between methods (at end) and results to understand.

The Discussion section reads more like a review article than a discussion of what this study has actually shown. It is interesting background, but not directly relevant to the experiments and analyses at hand. This section would benefit from less extraneous information and more direct discussion of what the present analyses have added to the field.

It would be nice to see examples of axonal activity so we can see what they are imaging earlier in the manuscript – they don't show an example until Figure 4.

Figure 5 was difficult to resolve and thus hard to evaluate.

*Reviewer #3 (Recommendations for the authors):*

Figure 3 resolution is very poor for some reason, and I was not able to evaluate the data presented.

---

## [Author Response]

Essential revisions:1) It will be essential to convince readers, including the editors and reviewers that the observed heterogeneity is not merely a reflection of motion artifacts. In discussion, it was recognized that this is difficult and might require new analyses or even new experiments, however it is a technical issue that goes to the heart of one of the main conclusions of the paper and so should be addressed rigorously.

Thank you for this suggestion. We addressed this concern by performing a new experiment using non-calcium-dependent mCherry expressed in cholinergic cortical axons. An example recording is included as a new supplemental video (Video 3) and data were analyzed in an identical manner to that of our experimental GCaMP6s data set. We did not find any evidence that motion of axons contributed to our results. We show that motion cannot account for our finding that cholinergic and noradrenergic axons increase activity around changes in behavioral state (i.e. whisking and walking; see Figure 2F). We also demonstrate that correlation between mCherry axons does not change at the onset of animal movement (see Figure 4 —figure supplement 1, panel G) or as a function of distance between axons (Figure 4L,P). We conclude that our experimental results were not impacted by movement of axons during our recordings. These findings are completely consistent with our previous results demonstrating that non-activity dependent fluorescent blebs also do not show movement-related changes in fluorescence. The addition of analysis of non-activity dependent axons facilitated a more thorough investigation of the partial correlational structure of GCaMP6s axons across the dorsal cortex.

2) Many details of the presentation were suboptimal and would benefit from revision. These include:a) Improved clarity in the description of the methods, for example in the methods used to separate sources of variance in figure 3 and elsewhere.b) Earlier and increased use of example recordings of single axon segment activity.

In the revised manuscript we have addressed these concerns by explaining more clearly the methods used, both experimentally and during analysis. We also now include example recordings in the manuscript to better illustrate that robustness of our results, even in raw recordings. The original submission included a raw example in Figure 1 and Figure 4 and a video of raw data collection as the second supplemental video. We have included histological verification of cholinergic and noradrenergic axons in the current submission (Figure 1 —figure supplement 1), an example of a typical field of view during axonal recordings (Figure 1 —figure supplement 1G), as well as a third supplementary video of raw mCherry images (Video 3). We feel that these examples are sufficient for readers to understand the nature of the data presented.

c) Improved reproduction of figure 5.

We apologize for the low quality reproduction of this figure in the original submission. This issue should be resolved in the current submission.

d) A discussion that more directly addresses the significance of the present findings.

The Discussion section has been edited considerably. We feel that the current submission is concise and focused on the presented results while providing necessary context.

Reviewer #1 (Recommendations for the authors):1) As stated before, the manuscript needs better controls for movement-induced artifacts as they might contribute to the heterogeneity in the recorded signals across axons. A control in which motion artifacts are quantified without a biased selection criterion is needed to better support the claim of heterogeneity. Ideally- Ca+ insensitive structures of the same size as the axonal ROIs might be already present in the recordings and can be selected with an unbiased criterion. Selecting particles of the same size and shape as axonal varicosities exhibiting both red and green fluorescence is a possibility, as GCaMP-expressing axons should not be red fluorescent. Adding some control animals expressing GFP instead GCaMP to the dataset and repeating the different metrics is also a possibility.

See above description of mCherry control experiment. We have now expressed non-activity-dependent mCherry in cholinergic axons, resulting in data on axons of the same shape and size as our experimental data set. These non-activity dependent axons do not exhibit significant motion artifacts and help us to interpret our activity-dependent axonal results.

2) While the prospect that the measured signals do not completely relate to action potential induced changes in axonal calcium is raised in lines 410-413, the manuscript does not explicitly address there or later in the discussion that the observed heterogeneity might arise from local, axon-specific presynaptic modulations and not because of differences in the spiking activity across axons (see for example Pardi et al. Science 2020). In the case the heterogeneity is not caused by movement-induced artifacts, these two alternative explanations should be explicitly addressed when interpreting the results.

The possibility of regional modulation of presynaptic terminals and axons is now mentioned in the Discussion section. Specifically, axon-specific presynaptic modulation is discussed in the current version of the manuscript.

3) Besides controlling for movement artifacts using an unbiased criterion for selecting calcium-insensitive structures, the manuscript would benefit with more extensive controls of the potential contribution of movement induced artifacts to the different metrics:a. When measuring the probability that a given axon will respond to a behavioral event (lines 191-201). What are the values for non-GCaMP (bleb), fluorescent particles?

We have included the distribution of the probability of mCherry fluorescence to meet our criteria for a response to behavioral events (supplement to Figure 2). These results reveal that the fluorescence of non-activity dependent axons is very unlikely to meet our criteria for responsiveness to a behavioral event.

b. How does correlation between Ca+-insensitive structures depend on distance (like in Figure 4J).

We analyzed the ordinary correlation and partial correlation between mCherry axons as a function of distance. Results are presented in Figure 4H,L,P. These results demonstrate that the correlations and partial correlations of non-activity dependent axons are not dependent upon distance.

4) While the examples in Figure 1 make very clear the existence of a global signals across regions and axons, none of the examples illustrates well the heterogeneity in the signals. Better exemplars showing the heterogeneity picked up by the analyses in Figure 4 would strengthen the claims.

We had addressed this concern in Figure 4 and Figure 2 —figure supplement 1. These figures now form the basis of our evidence for heterogeneity in the axonal activity.

5) Line 1006. What criteria is used to determine if an ROI corresponds to an axon? Are most ROIs on varicosities as stated in line 1012 when describing blebs, or on axonal shafts as in the examples on Figure 4A? If they are mostly on varicosities, analyses from same axon pairs should also come from ROIs of the same size and shape as the rest of the data, as the size and shape might influence correlation values.

The manuscript now clarifies that ROIs were taken from axonal shafts. Our new control data set using cholinergic axons expressing non-activity dependent mCherry consists of ROIs that are similar in size and shape to our experimental GCaMP6s data set.

6) What constitutes a 'bleb"? How are they distinguished from regular varicosities besides the lack of calcium-dependent activity? What is their size and shape compared to regular axonal ROIs?

This reviewer makes an excellent point that blebs do not match the size and shape of axons used in the experimental data set, and that there could be bias when selecting for non-calcium-dependent fluorescence. For this and other reasons, we feel that the new mCherry data set is a much stronger control than blebs. We continue to present the data on non-activity dependent blebs in our current manuscript, as data supporting our results obtained with the more vigorous control of non-activity dependent mCherry labeled cholinergic axons.

7) Histological analysis of the GCaMP6 labelling is needed to control if the intended structures are the only ones labelled with GCaMP in both groups.

The current submission includes histological confirmation that all, or nearly all, of our GCaMP6+ axons were indeed cholinergic or noradrenergic (antibody positive for the presence of ChAT or TH) (Figure 1 —figure supplement 1).

8) The original reference of the axon-GCaMP sensor used should be cited, not only the supplier of the virus (line 943).

The original reference (Broussard et al., 2018) is now included in the methods.

9) Figure 3B does not have a time scale.

This oversight has been corrected in the current submission.

10) It is better to avoid using non metric units (line 952).

The use of fahrenheit has been changed to celsius.

Reviewer #2 (Recommendations for the authors):The authors miss the opportunity to compare behavioral state modulation across vs within cortical regions – that is, how similar or different is the "common signal" between different regions of cortex? They have these data (an advantage of the mesoscopic scale), and such an analysis would add significantly to the impact of the study.

The data presented in this paper have not been rigorously aligned to a cortical area map. While rough alignment to the Allen Institute Common Coordinate Framework was performed to create Figure 1B using large landmarks (i.e. midline, frontal poles, and/or λ), this alignment is not precise enough to make claims about differences within vs. between cortical areas (except in broad terms). We believe that to make this comparison would require functional mapping of all of the areas examined in each mouse. This precise mapping within individual mice was beyond the scope of the present study.

To approximate such a comparison we have presented correlations as a function of distance. It is reasonable to assume that axons within the same ROI (presented as 0 mm distance in Figure 4) are from the same region. As distance increases, so does the likelihood that the axons have been sampled from different regions of the cortex. We agree that further investigation of the strength of the common signal in different cortical areas would be interesting and impactful.

The Results section is high in jargon and short on technical details related to the analyses shown, which are necessary for the reader to fully understand the meaning and potential significance of the analysis. This section would benefit from a clearer narrative about what exactly each analysis adds to the overall picture so the reader does not have to keep toggling between methods (at end) and results to understand.

The results have been substantially edited, both for clarity and organization. We have tried to reduce the jargon and make a cleaner presentation. We hope that the current form is easier to follow.

The Discussion section reads more like a review article than a discussion of what this study has actually shown. It is interesting background, but not directly relevant to the experiments and analyses at hand. This section would benefit from less extraneous information and more direct discussion of what the present analyses have added to the field.

The Discussion section has been substantially edited for conciseness and clarity.

It would be nice to see examples of axonal activity so we can see what they are imaging earlier in the manuscript – they don't show an example until Figure 4.

Raw axonal activity is illustrated in Figures 1 and 4, Figure 2 —figure supplement 1, and in Videos 2, 3. We feel that these are sufficient for the reader to understand the nature of our data early in reading the manuscript.

Figure 5 was difficult to resolve and thus hard to evaluate.

This issue should be remedied in the current submission.

Reviewer #3 (Recommendations for the authors):Figure 3 resolution is very poor for some reason, and I was not able to evaluate the data presented.

This has been corrected.